# LONP1 and mtHSP70 cooperate to promote mitochondrial protein folding

Chun-Shik Shin [1], Shuxia Meng[1], Spiros D. Garbis[2], Annie Moradian [2,4], Robert W. Taylor [3], Michael J. Sweredoski [2,5], Brett Lomenick[2] & David C. Chan[1✉]

Most mitochondrial precursor polypeptides are imported from the cytosol into the mitochondrion, where they must efficiently undergo folding. Mitochondrial precursors are imported as unfolded polypeptides. For proteins of the mitochondrial matrix and inner membrane, two separate chaperone systems, HSP60 and mitochondrial HSP70 (mtHSP70), facilitate protein folding. We show that LONP1, an AAA+ protease of the mitochondrial matrix, works with the mtHSP70 chaperone system to promote mitochondrial protein folding. Inhibition of LONP1 results in aggregation of a protein subset similar to that caused by knockdown of DNAJA3, a co-chaperone of mtHSP70. LONP1 is required for DNAJA3 and mtHSP70 solubility, and its ATPase, but not its protease activity, is required for this function. In vitro, LONP1 shows an intrinsic chaperone-like activity and collaborates with mtHSP70 to stabilize a folding intermediate of OXA1L. Our results identify LONP1 as a critical factor in the mtHSP70 folding pathway and demonstrate its proposed chaperone activity.

[1] Division of Biology and Biological Engineering, California Institute of Technology, Pasadena, CA 91125, USA. [2] Proteome Exploration Laboratory of the Beckman Institute, California Institute of Technology, Pasadena, CA 91125, USA. [3] Wellcome Centre for Mitochondrial Research, Translational and Clinical Research Institute, The Medical School, Newcastle University, Newcastle upon Tyne NE2 4HH, UK. [4] Present address: Cedars-Sinai Medical Center, Los Angeles, CA, USA. [5] Present address: Kaiser Permanente, Los Angeles, CA, USA. ✉email: dchan@caltech.edu

Although human mitochondria contain their own genome encoding 13 core subunits of the oxidative phosphorylation (OXPHOS) machinery, the majority of the human mitochondrial proteome is encoded by the nuclear genome[1,2]. After synthesis on cytosolic ribosomes, nuclearly encoded mitochondrial precursors are imported into the organelle in their unfolded state due to the limited pore size of the import machinery[3]. Therefore, after import, mitochondrial precursors must efficiently fold into their functional structure to avoid aggregation due to exposure of hydrophobic surfaces. In the mitochondrial matrix, two highly conserved chaperone systems, HSP60 and mitochondrial HSP70 (mtHSP70), are critical in facilitating the folding reaction of mitochondrial precursors[4,5]. Mammalian HSP60 is related to the bacterial chaperonin GroEL and forms a tetradecameric (double-ringed heptamer) protein complex that encapsulates unfolded polypeptide chains between 20 and 50 kDa in size to accelerate substrate folding. In yeast, Hsp60 is an essential protein required for mitochondrial protein folding and biogenesis[6].

mtHSP70 is also essential for mitochondrial protein biogenesis, with functions in both mitochondrial precursor import and folding[7]. As the key component of the presequence translocase-associated motor (PAM) of the inner membrane, mtHSP70 uses ATP hydrolysis to drive precursor import into the matrix. Subsequently, as part of a chaperone complex with the soluble J-protein DNAJA3, and a nucleotide exchange factor (either GRPEL1 or GRPEL2), it promotes the folding of newly imported polypeptides into their native structures[8]. Therefore, these two distinct mtHSP70 complexes function sequentially for mitochondrial protein biogenesis in the matrix, firstly to drive directional translocation of mitochondrial precursors and subsequently to facilitate their folding[9]. As an ATP hydrolyzing enzyme, mtHSP70 cycles between ATP and ADP-bound forms. ATP-bound mtHSP70 is an "open" form that shows rapid binding and release of its substrates. ATP hydrolysis, enhanced by the J-protein, results in a 'closed' ADP-bound mtHSP70 that stably interacts with substrate. ADP-ATP exchange, induced by GRPEL1 or GRPEL2, causes substrate release and a new cycle of 'open' mtHSP70[4,7].

Another mitochondrial protein with potential chaperone function is LONP1, an AAA + (ATPase associated with a wide variety of cellular activities) protease of the mitochondrial matrix[10,11]. LONP1 contains 3 functional domains: an N-terminal domain responsible for substrate recognition, an ATP-binding and hydrolyzing AAA+ ATPase domain, and a C-terminal protease domain. It assembles into a hexameric cylindrical chamber for substrate protein unfolding and proteolysis. After entry of misfolded or damaged substrates, ATP hydrolysis drives substrate unfolding and translocation to the proteolytic chamber. Known degradation substrates of LONP1 include oxidized mitochondrial aconitase[12], the unassembled alpha-subunit of the mitochondrial processing peptidase[13], and DNA-free TFAM[14]. In addition to this well-established protease function, LONP1 has been proposed to have a chaperone-like function in the assembly of oxidative phosphorylation (OXPHOS) complexes in yeast and mammalian mitochondria, independently of its protease activity[15,16]. This proposed function is based on the observation that loss of LONP1 results in reduced levels of assembled OXPHOS complexes, a defect that can be rescued by a protease-deficient mutant of LONP1. However, the molecular basis of this effect is poorly defined, and there has been no direct demonstration that LONP1 has intrinsic chaperone activity on a substrate. In this study, we show that LONP1 plays a major role in mitochondrial protein biogenesis. In a reconstituted in vitro assay, LONP1 shows not only an intrinsic chaperone activity but also synergistic cooperation with the mtHSP70 chaperone.

## Results

**LONP1 in mitochondrial protein biogenesis.** Knockdown of human mitochondrial Lon protease or its inhibition by the synthetic triterpenoid, CDDO, has been shown to cause the accumulation of electron dense aggregates within mitochondria[17–19]. To better understand the molecular basis for this protein aggregation, we tested the solubility of a panel of mitochondrial proteins with Triton X-100 extraction upon LONP1 inhibition (Fig. 1a). After LONP1 knockdown (Fig. 1b) or CDDO treatment (Fig. 1c), a substantial portion of OXA1L and NDUFA9 partitioned into the detergent-insoluble fraction. We tested these proteins because they have important functions in mitochondrial biology. OXA1L is an insertase of the inner membrane and interacts with mitochondrial ribosomes during the biogenesis of mtDNA-encoded proteins. NDUFA9 is a subunit of the respiratory chain Complex I (NADH dehydrogenase). In both cases, the insoluble proteins were in their mature forms that lacked the mitochondrial presequence, indicating successful import into the matrix and presequence cleavage. VDAC1, an outer membrane protein, remained soluble upon LONP1 knockdown or CDDO treatment. We confirmed that there was no protein loss during our fractionation process (Supplementary Fig. 1a). In the mitochondrial matrix, CLPXP is a related ATP-dependent protease that is a complex between the protease CLPP and the AAA+ ATPase CLPX. Knockdown of CLPP or CLPX did not affect OXA1L and NDUFA9, suggesting that their solubility is specifically sensitive to LONP1 inhibition (Supplementary Fig. 1b).

Given the proposed function of LONP1 as a chaperone and protease, we addressed whether protein aggregation upon LONP1 inactivation resulted from a failure of protein degradation or a failure of protein biogenesis (import and folding). To address this issue, we developed a doxycycline (DOX)-inducible system to express siRNA-resistant LONP1 and titrated DOX concentrations to control its expression level. The first possibility was excluded on account of a protease-deficient LONP1[S855A] mutant[20] being effective at restoring the solubility of OXA1L and NDUFA9 in LONP1 knockdown cells when induced with 0.05 μg/ml DOX, which resulted in endogenous levels of LONP1[S855A] (Fig. 1b). It should be noted that although DOX in the micromolar range inhibits mitochondrial function, concentrations up to 0.1 μg/ml do not disrupt mitochondrial translation (Supplementary Fig. 1c), respiration, or cell growth[21,22]. To examine the second possibility, we asked whether vulnerable proteins, like OXA1L and NDUFA9, aggregated only during their biogenesis or throughout their lifecycle. To prevent new protein biogenesis, we treated cells with cycloheximide (CHX) to inhibit cytosolic translation. CHX blocked the ability of CDDO to cause OXA1L or NDUFA9 insolubility (Fig. 1c). Moreover, when cells were treated with the protonophore CCCP to inhibit mitochondrial matrix import of precursors, OXA1L and NDUFA9 insolubility were also prevented (Fig. 1c). Finally, inhibition of mitochondrial import into the matrix using TIM44 knockdown also prevented protein insolubility caused by CDDO and LONP1 knockdown (Fig. 1c and Supplementary Fig. 1d). These results indicate that mature, folded OXA1L and NDUFA9 are stable upon LONP1 inhibition and suggest that these proteins are vulnerable to aggregation only during their biogenesis phase.

To directly test the temporal characteristics of protein aggregation, we generated a DOX-inducible expression system for OXA1L-FLAG. At 0.1 μg/ml DOX, which does not affect mitochondrial translation (Supplementary Fig. 1c)[21,22], OXA1L-FLAG produced in this system was detected in the detergent-soluble fraction of mitochondria (Fig. 1d and Supplementary Fig. 1e). CDDO treatment resulted in substantial insolubility in detergent (Fig. 1d) and coalescence of OXA1L into punctate aggregates within mitochondria (Fig. 1e). To determine whether

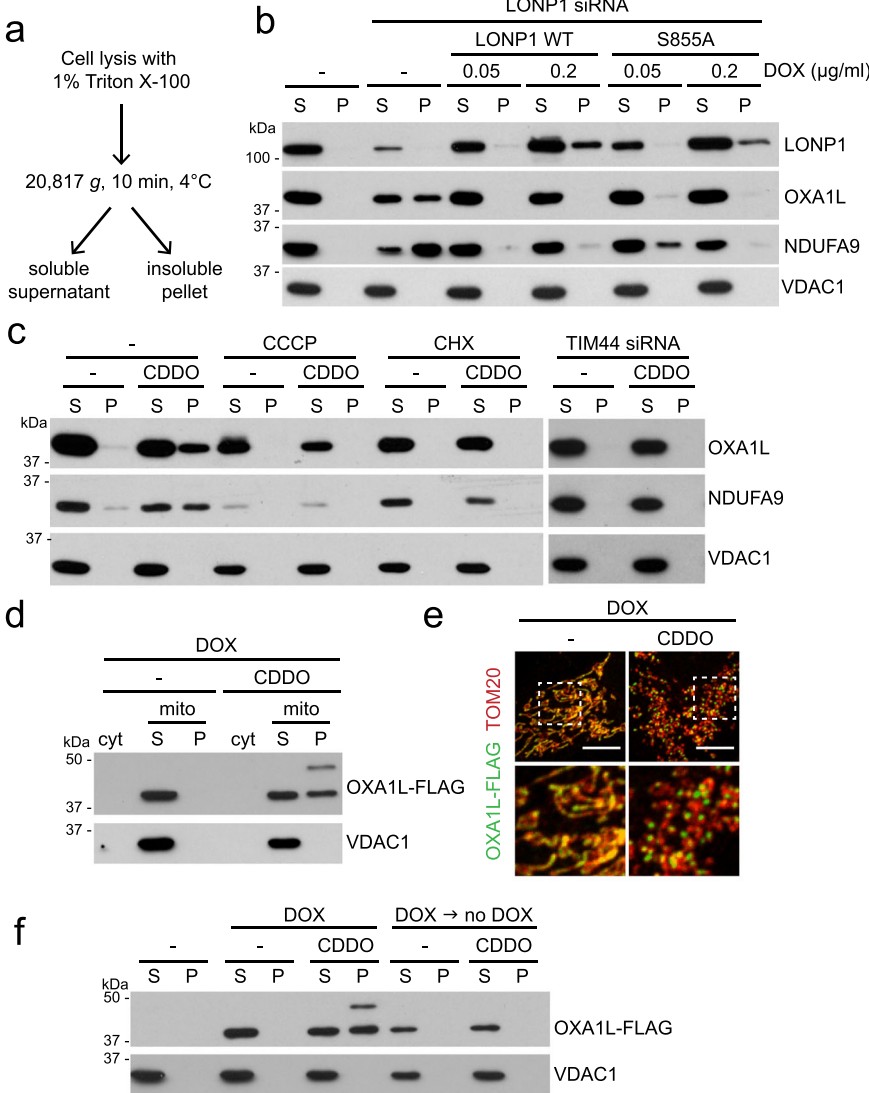

**Fig. 1 LONP1 required for OXA1L and NDUFA9 biogenesis. a** Schematic of Triton X-100 extraction to determine protein solubility. **b** Protein aggregation upon LONP1 knockdown and its rescue by protease-deficient LONP1. Triton X-100 extraction and Western blotting were used to examine protein solubility in LONP1 knockdown 143B cells re-expressing doxycycline (DOX)-inducible RNAi-resistant LONP1[WT] and protease-deficient LONP1[S855A]. **c** Protein insolubility with CDDO treatment, and its rescue by drug treatment or TIM44 knockdown. Triton X-100 extraction and Western blotting were used to examine protein solubility in 143B cells treated with vehicle (-) or CDDO. Additional drug treatment (CCCP or CHX) or TIM44 knockdown was performed as indicated. **d** Solubility of OXA1L-FLAG. Mitochondria were isolated from 143B cells expressing doxycycline (DOX)-inducible OXA1L-FLAG and analyzed by Triton X-100 extraction and Western blotting. Expression was induced by DOX for 16 h in the absence or presence of 2 μM CDDO. cyt, cytosol; mito, mitochondria. **e** Visualization of OXA1L-FLAG aggregation. OXA1L-FLAG was induced by DOX in the absence or presence of 2 μM CDDO, and visualized by anti-FLAG immunostaining. Tom20 was used as a mitochondrial marker. Scale bar, 10 μm. **f** Temporal features of OXA1L aggregation. Triton X-100 extraction and Western blotting were used to examine OXA1L-FLAG solubility. Cells were treated with DOX and 2 μM CDDO for 24 h simultaneously or treated with DOX for 24 h, and after DOX removal, treated with 2 μM CDDO for an additional 24 h.

mature, folded OXA1L-FLAG was vulnerable to LONP1 inhibition, we used DOX treatment to induce a discrete pulse of OXA1L-FLAG. DOX was then removed to shut down the expression of OXA1L-FLAG, and the cells were treated with CDDO. With this experimental sequence, OXA1L-FLAG maintained its solubility in the face of CDDO treatment (Fig. 1f). These results further suggest that pre-existing OXA1L is stable to LONP1 inhibition, and that OXA1L aggregates only when LONP1 is blocked during its biogenesis phase.

**LONP1 intersects with the mtHSP70 pathway.** Because depletion of LONP1 results in OXA1L and NDUFA9 aggregation, it is

possible that LONP1 depletion disrupts the chaperone systems that facilitate protein folding in the mitochondrial matrix. We therefore tested the status of OXA1L upon disruption of HSP60 and mtHSP70 (also known as HSPA9), the two major chaperone systems of the matrix[4]. HSP60 knockdown had no effect on solubility of OXA1L (Supplementary Fig. 2a). As might be expected, knockdown of mtHSP70 was problematic to interpret, because it greatly reduced the steady-state level of OXA1L and other mitochondrial proteins (Supplementary Fig. 2a). This result probably reflects the additional, known role of mtHSP70 as the import motor of the translocase of the inner membrane (TIM)[7,8].

To preserve the import function of mtHSP70 and selectively inhibit its folding function, we knocked down DNAJA3 (also

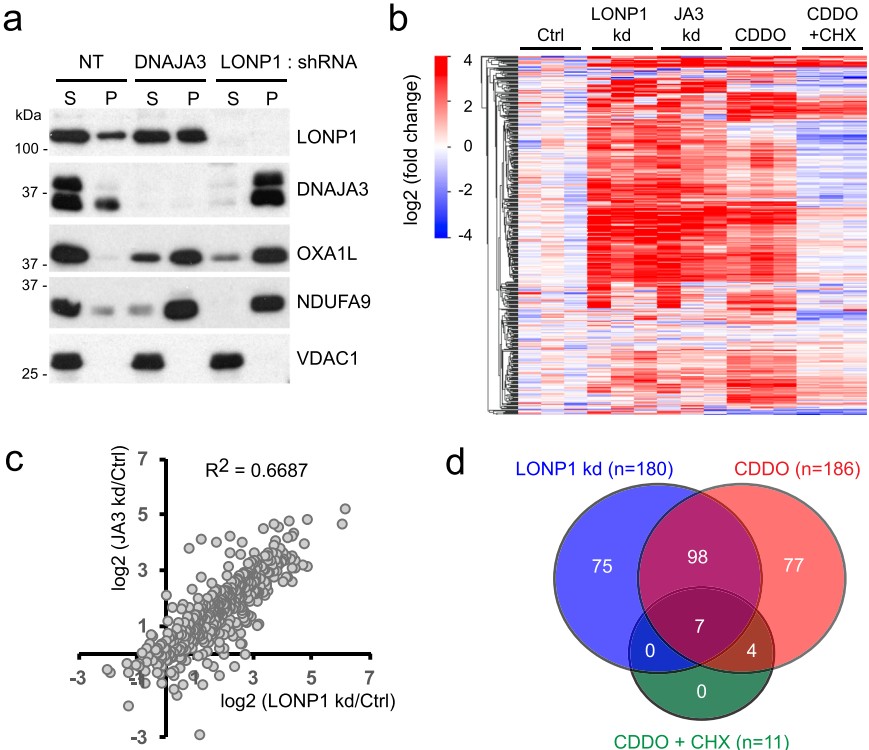

**Fig. 2 Aggregation profile of mitochondrial proteins in LONP1 or DNAJA3 knockdown and CDDO-treated cells. a** Comparison of mitochondrial protein solubility in DNAJA3 and LONP1 knockdown cells. Cells were analyzed by Triton X-100 extraction and Western blotting. Note that the anti-DNAJA3 antibody recognizes a doublet, with both bands disappearing after DNAJA3 knockdown. NT: non-target. **b, c** Comparison of mass spectrometry data from three independent LONP1 knockdown, DNAJA3 (JA3) knockdown, CDDO alone and CDDO + cycloheximide (CHX)-treated mitochondria. LC-MS was used to identify proteins within detergent-insoluble pellets isolated from mitochondria. Clustered heatmap (**b**) and scatter plot (**c**) show the fold-change (compared to Ctrl) in insolubility of 445 mitochondrial proteins. **d** Venn diagram analysis of significantly aggregated proteins (at least 2-fold over control with a *p*-value <0.05) in LONP1-knockdown, CDDO alone and CDDO + CHX-treated mitochondria. Statistical analysis of proteins enriched in mitochondrial aggregates was performed using the LIMMA moderated two-tailed *t*-test.

known as TID1), a J domain protein that acts as a co-chaperone for the folding function of mtHSP70 by enhancing its ATP hydrolysis activity[23]. In yeast, the DNAJA3 orthologue Mdj1 similarly functions as a co-chaperone of mtHSP70 during mitochondrial precursor folding[9]. In multiple human cell lines, we found that knockdown of DNAJA3 severely inhibited the solubility of OXA1L and NDUFA9 (Fig. 2a and Supplementary Fig. 2b) without affecting their steady-state levels. This observation is consistent with a previous report showing DNAJA3 knockdown causes electron-dense aggregates inside mitochondria[24].

To assess the full range of protein insolubility in LONP1 and DNAJA3 knockdown cells, we performed liquid chromatography-tandem mass spectrometry (LC-MS) analysis of mitochondrial, detergent-insoluble pellets. OXA1L (3.0 and 5.8-fold) and NDUFA9 (3.2 and 3.7-fold) were significantly enriched in insoluble pellets from LONP1 and DNAJA3 knockdown mitochondria, respectively (Supplementary Data 1). We found that the aggregation profiles in these two knockdown conditions were highly correlated with a correlation coefficient $R^2$ of 0.6687 (Fig. 2b, c). 135 mitochondrial proteins (about 10% of the human mitochondrial proteome) were aggregated more than 2-fold with a significant *p*-value (<0.05) in both LONP1 and DNAJA3 knockdown mitochondria compared to control mitochondria (Supplementary Fig. 2c and Supplementary Data 2). The vast majority of these aggregated proteins (130 out of 135 proteins) are known to localize to the mitochondrial matrix or mitochondrial inner membrane[25] (Supplementary Fig. 2d). These results indicate that LONP1 and the mtHSP70-DNAJA3 chaperone system function in

the stabilization of a very similar and large set of mitochondrial proteins.

We performed additional proteomic analysis of mitochondrial aggregates from CDDO-treated versus CDDO + CHX-treated cells to distinguish between proteins that require LONP1 function during their biogenesis versus those that are aggregation-prone even after folding. CDDO treatment caused an aggregation profile highly similar to LONP1 knockdown (Fig. 2b, d). Some examples include UQCRC1 (a subunit of respiratory chain Complex III), MRPL23 (a subunit of mitochondrial ribosome), and CLPX (a subunit of the AAA+ protease CLPXP) (Supplementary Fig. 2e). These proteins were enriched in mitochondrial aggregates from cells treated with LONP1 shRNA, DNAJA3 shRNA, or CDDO. However, these proteins were not enriched in mitochondrial pellets from cells co-treated with CDDO and CHX. Out of 186 proteins enriched in mitochondrial aggregates in CDDO-treated cells, only 11 remained aggregated with CHX co-treatment (Fig. 2d).

**Depletion of LONP1 results in mtHSP70 aggregation.** Consistent with a functional connection between LONP1 and the mtHSP70 chaperone system, DNAJA3 and mtHSP70 were highly insoluble in cells containing shRNA or siRNA against LONP1 (Figs. 2a, 3a) or cells treated with CDDO (Supplementary Fig. 3a). GRPEL1, the nucleotide exchange factor for mtHSP70 in human mitochondria[26], was only slightly affected (Supplementary Fig. 3b). Although CDDO is known to have targets other than LONP1, the aggregation of mtHSP70 and DNAJA3 by CDDO

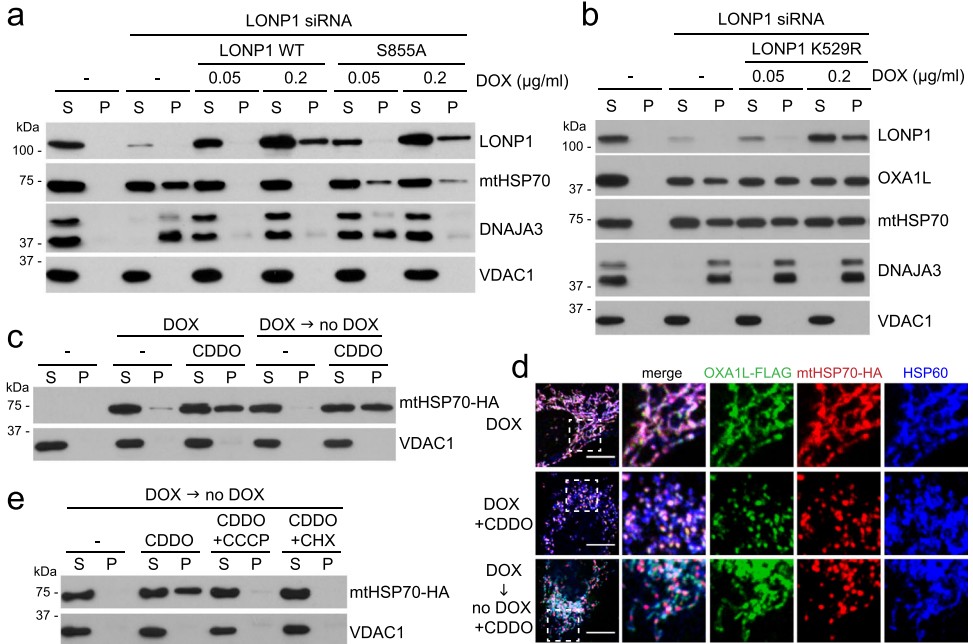

**Fig. 3 mtHSP70 aggregation with LONP1 depletion. a** Rescue of mtHSP70 solubility with protease-deficient LONP1. Triton X-100 extraction and Western blotting were used to analyze protein solubility in LONP1 knockdown cells re-expressing doxycycline (DOX)-inducible RNAi-resistant LONP1^WT and protease-deficient LONP1^S855A. **b** Protein aggregation in LONP1 knockdown cells re-expressing ATPase-deficient LONP1. Triton X-100 extraction and Western blotting were used to examine protein solubility in LONP1 knockdown 143B cells re-expressing doxycycline (DOX)-inducible RNAi-resistant LONP1^K529R. **c** Temporal features of mtHSP70 insolubility. mtHSP70 solubility was analyzed in 143B cells expressing DOX-inducible mtHSP70-HA. Cells were treated with DOX and 2 μM CDDO simultaneously or sequentially, as in Fig. 1f. **d** Comparison of OXA1L versus mtHSP70 solubility. 143B cells expressing DOX-inducible OXA1L-FLAG and mtHSP70-HA were treated with DOX and CDDO simultaneously or sequentially. Note the aggregation of mtHSP70 but not OXA1L in the sequential treatment. Scale bar, 10 μm. **e** Rescue of CDDO-induced mtHSP70 insolubility by CCCP or CHX. mtHSP70 solubility was analyzed in 143B cells expressing DOX-inducible mtHSP70-HA. Cells were treated with DOX and CDDO sequentially. CCCP or CHX was co-treated with CDDO as indicated.

treatment was efficiently prevented by re-expression of the protease-deficient LONP1^S855A mutant[20] (Supplementary Fig. 3a), indicating that the aggregation effect is specific to LONP1 inhibition. Similarly, aggregation caused by LONP1 knockdown was rescued by LONP1^S855A (Fig. 3a). In contrast, a ATPase-deficient LONP1^K529R mutant[20] did not rescue the aggregation of mtHSP70, DNAJA3, and OXA1L in LONP1 knockdown cells (Fig. 3b) and in fact acted dominantly to promote protein aggregation in WT cells (Supplementary Fig. 3c). Taken together, these results indicate that key components of the mtHSP70 system aggregate in the absence of LONP1.

We generated a DOX-inducible expression system for mtHSP70-HA to test whether LONP1 is required only during the protein's biogenesis phase, as was the case for OXA1L. Like endogenous mtHSP70, a substantial fraction of mtHSP70-HA was insoluble in the presence of CDDO (Supplementary Fig. 3d). In contrast to OXA1L, however, CDDO added after a discrete pulse of mtHSP70-HA induction still caused insolubility (Fig. 3c). This observation indicates that mature mtHSP70, unlike mature OXA1L, is vulnerable to LONP1 inactivation. To visualize their different vulnerabilities, we used DOX to simultaneously induce expression of both OXA1L-FLAG and mtHSP70-HA. With ongoing DOX induction, CDDO caused both proteins to aggregate into co-localizing, punctate spots along mitochondria. When CDDO was added subsequent to a pulse of DOX, mtHSP70-HA but not OXA1L-FLAG formed aggregated structures (Fig. 3d). These biochemical and imaging results indicate that mtHSP70 and OXA1L have distinct temporal requirements for LONP1.

Aggregation of preformed mtHSP70-HA could be prevented by TIM44 knockdown (Supplementary Fig. 3e, f) or treatment with either CCCP or CHX (Fig. 3e). Each of these treatments prevents the import of mitochondrial precursors. Taken together, these results suggest that mature mtHSP70 aggregates upon LONP1 inhibition, unless mitochondrial import is blocked.

**LONP1 co-immunoprecipitates with mtHSP70.** Consistent with the functional interplay between LONP1 and mtHSP70, we found that the two proteins co-immunoprecipitated (Fig. 4a and Supplementary Fig. 4a). This co-immunoprecipitation was strongly enhanced by knockdown of GRPEL1. This result suggests that the interaction between LONP1 and mtHSP70 is regulated by the latter's adenine nucleotide state. The interaction of DNAJA3 with mtHSP70 was also dramatically increased under GRPEL1 knockdown conditions (Fig. 4a). Analogously, in yeast, mtHSP70 dissociates from the DnaJ protein Mdj1 much more slowly in the absence of a nucleotide exchange factor[27].

Because GRPEL1 knockdown is expected to favor occupancy of mtHSP70 with ADP, we tested LONP1-mtHSP70 interactions under culture conditions that deplete intracellular ATP. Cells cultured in high glucose generate ATP from both cytosolic glycolysis and mitochondrial oxidative phosphorylation, whereas cells grown in galactose primarily depend on oxidative phosphorylation[28]. Consistent with this concept, the mitochondrial ATP synthase inhibitor oligomycin greatly reduced the cellular ATP level in galactose-fed 143B cells but not glucose-fed cells (Fig. 4b). Under such ATP-depleted conditions, we found enhanced co-imunoprecipitation of mtHSP70 with LONP1 and DNAJA3 (Fig. 4c and Supplementary Fig. 4b).

We tested whether pathogenic *LONP1* variants associated with human mitochondrial disease cause mitochondrial protein

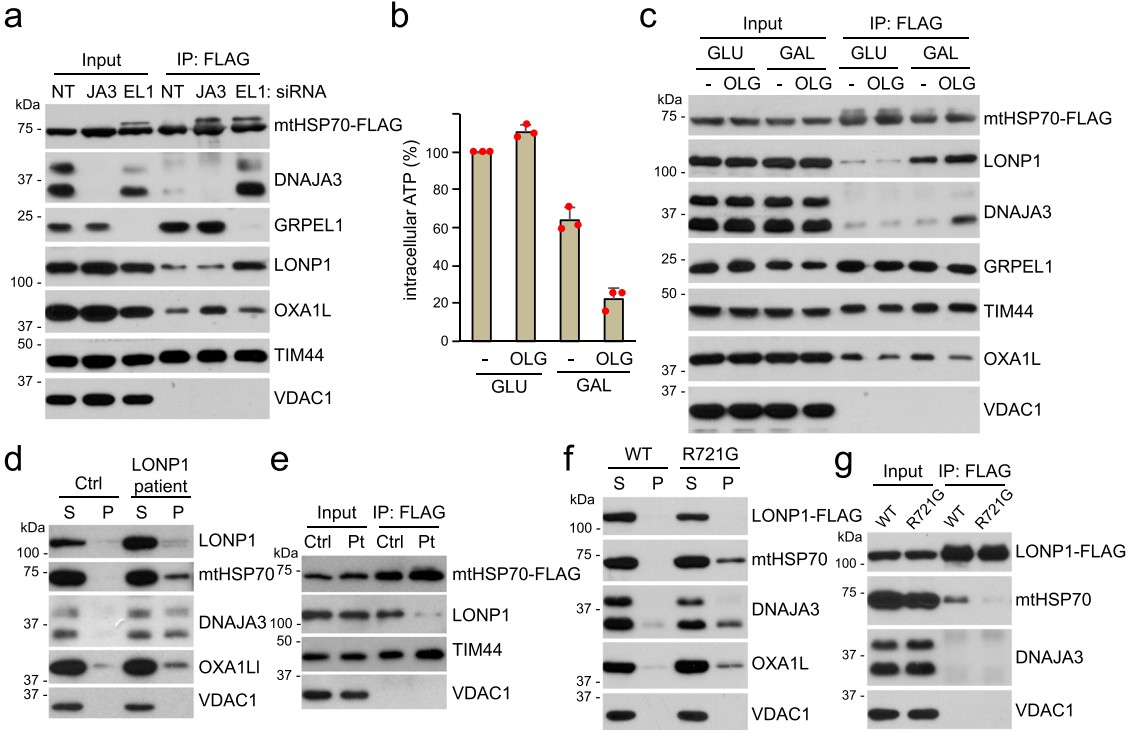

**Fig. 4 Co-immunoprecipitation of LONP1 and mtHSP70. a** Physical interactions of mtHSP70-FLAG upon DNAJA3 (JA3) or GRPEL1 (EL1) knockdown. mtHSP70-FLAG was constitutively expressed, immunoprecipitated and analyzed by Western blotting. NT: non-target. **b** Quantification of intracellular ATP after glucose (GLU) or galactose (GAL) culture in the absence or presence of 5 μM oligomycin (OLG) for 2 h. Red dots indicate individual data points ($n = 3$ independent experiments). Data are presented as means ± standard deviation. **c** Effect of galactose culture and oligomycin treatment on mtHSP70 interactions. Immunoprecipitates of mtHSP70-FLAG were analyzed for the indicated proteins. **d** Mitochondrial protein solubility in control (Ctrl) and LONP1 patient (bi-allelic p.Tyr565His and p.Glu733Lys variants) fibroblasts. **e** mtHSP70 interactions in control (Ctrl) and LONP1 patient (Pt) fibroblast cells. mtHSP70-FLAG was expressed in the indicated cells and immunoprecipitated. **f** Comparison of LONP1[WT] and CODAS LONP1[R721G] in rescuing mitochondrial protein aggregation in LONP1 knockdown cells. 143B cells expressing siRNA-resistant LONP1[WT] and CODAS LONP1[R721G] were treated with siRNA against LONP1 and analyzed by Triton X-100 extraction. **g** Comparison of LONP1[WT] and CODAS LONP1[R721G] in interacting with mtHSP70. FLAG-tagged LONP1[WT] and CODAS LONP1[R721G] were expressed in 143B cells, and FLAG immunoprecipitates were analyzed for the indicated proteins.

aggregation. Using primary fibroblasts containing bi-allelic mutations (p.Tyr565His and p.Glu733Lys; NM_004793.3) within the LONP1 ATPase domain associated with congenital lactic acidosis, muscle weakness, and multiple mitochondrial OXPHOS deficiencies[29], we found substantially reduced solubility of mtHSP70, DNAJA3 and OXA1L (Fig. 4d). We expressed FLAG-tagged mtHSP70 in patient fibroblasts and tested whether the pathogenic LONP1 variants affected the co-imunoprecipitation of mutant LONP1 with mtHSP70. The LONP1-mtHSP70 interaction in patient cells was much reduced compared to control cells (Fig. 4e). We tested another pathogenic LONP1 variant (p. Arg721Gly; NM_004793.3) reported to cause CODAS (a syndrome characterized by cerebral, ocular, dental, auricular and skeletal abnormalities)[30] but because patient fibroblasts were not available, we tested the function of this mutant in cells knocked down for endogenous LONP1. The CODAS LONP1 mutant was much less effective than WT LONP1 in rescuing the protein aggregation defect (Fig. 4f). This mutant also showed reduced co-imunoprecipitation with mtHSP70 (Fig. 4g). These results suggest that pathogenic LONP1 mutants are defective in co-immunoprecipitation with mtHSP70 and have impaired protein stabilization function.

**Reconstitution of OXA1L solubilization.** Based on the results above, LONP1 appears to be important for stabilization of proteins like OXA1L during their biogenesis phase. An important issue is whether LONP1 has inherent chaperone activity, or

whether mtHSP70 is the true chaperone and LONP1 is simply necessary for mtHSP70 stability. To address the potential chaperone activity of LONP1, we developed an in vitro assay to reconstitute solubilization of OXA1L using recombinant proteins. OXA1L was produced by in vitro translation using purified, recombinant components[31] (Fig. 5a). In the absence of additional components, OXA1L produced in this way was quantitatively sedimented by a high speed spin, suggesting its aggregation (Fig. 5b). This result is expected, because the reaction contained neither a chaperone nor a biological membrane. OXA1L, a transmembrane protein of the inner membrane, would not be able to fold into its native structure.

Addition of WT LONP1 resulted in degradation of OXA1L, likely because it is recognized as a misfolded substrate (Supplementary Fig. 5a). This degradation did not occur with the protease-deficient LONP1[S855A] mutant. Addition of recombinant LONP1[S855A], but not ATPase-deficient LONP1[K529R], resulted in a small amount of OXA1L solubilization (Fig. 5a, b). In contrast, mtHSP70 without LONP1 was ineffective at solubilizing OXA1L, even when combined with its partners GRPEL1 and DNAJA3 (Fig. 5c, d). Instead, substantial fractions of all 3 proteins, especially DNAJA3, became insoluble along with OXA1L. In contrast, expression of the soluble, control protein DHFR did not affect their solubility (Supplementary Fig. 5b). These results suggest that the mtHSP70 chaperone complex is susceptible to aggregation when futilely engaged with a difficult folding substrate. These observations are also consistent with cell culture data showing that mtHSP70 and DNAJA3 are aggregated

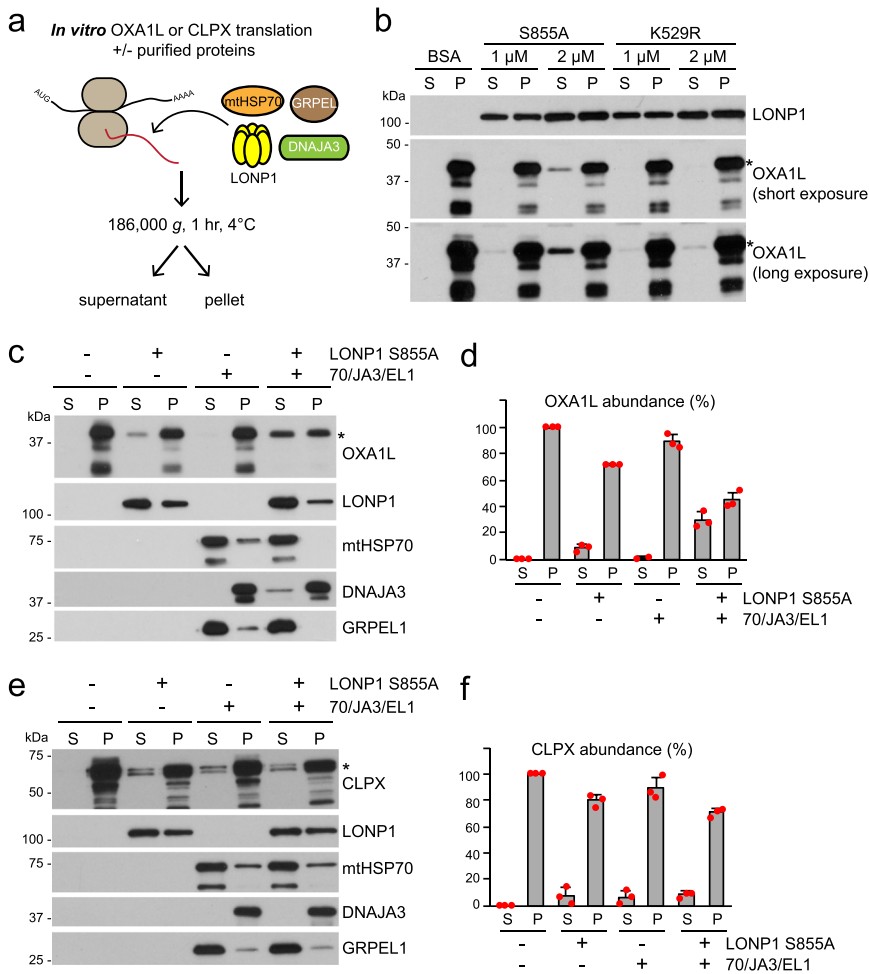

**Fig. 5 In vitro reconstitution of OXA1L and CLPX solubilization. a** Schematic of in vitro translation in the presence of recombinant LONP1 and mtHSP70 chaperone components. **b** Analysis of OXA1L solubility in the presence of protease-deficient LONP1$^{S855A}$ or ATPase-deficient LONP1$^{K529R}$. An asterisk (*) indicates full-length OXA1L. **c–f** Analysis of OXA1L and CLPX solubility in the presence of LONP1$^{S855A}$ plus mtHSP70 chaperone components (4 μM mtHSP70, 1 μM DNAJA3, 1 μM GRPEL1). **c** Analysis of OXA1L solubilization in vitro. Western blot was used to monitor solubility of the indicated components. **d** Quantification of OXA1L solubilization from three independent in vitro reactions. Red dots indicate individual data points (n = 3 independent experiments). Data are presented as means ± standard deviation. **e** Analysis of CLPX solubilization in vitro. Western blot was used to monitor the solubility of the indicated components. An asterisk (*) indicates full-length CLPX. **f** Quantification of CLPX solubilization from three independent in vitro CLPX reactions. Red dots indicate individual data points (n = 3 independent experiments). Data are presented as means ± standard deviation.

upon LONP1 inhibition unless mitochondrial import is disrupted (Fig. 3e and Supplementary Fig. 3e and f).

Addition of both LONP1$^{S855A}$ and mtHSP70 to the reaction increased solubility of OXA1L (Supplementary Fig. 5c). The highest activity was observed when LONP1 and mtHSP70 were supplemented with GRPEL1 and DNAJA3 together (Fig. 5c, d and Supplementary Fig. 5c) whereas individual addition of GRPEL1 or DNAJA3 with mtHSP70 did not improve OXA1L solubility (Supplementary Fig. 5d). In such reactions, LONP1 protected the mtHSP70 chaperone components against OXA1L-induced aggregation. The DNAJA3 and GRPEL1 co-chaperones increased solubilization of OXA1L in the presence of mtHSP70 and LONP1, but had no activity without mtHSP70 (Supplementary Fig. 5c). Consistent with the cell culture results, the ability of LONP1$^{S855A}$ and the mtHSP70 chaperone in solubilizing OXA1L was substantially disrupted by CDDO (Supplementary Fig. 5c). Because there are no biological membranes in this in vitro system, it is unlikely that the solubilized OXA1L can fold into its native, membrane-embedded state. It therefore seemed likely that solubilized OXA1L would need to be stabilized by binding to

LONP1 and mtHSP70 chaperone components. Immunoprecipitates of soluble OXA1L indeed contained LONP1$^{S855A}$, mtHSP70, and DNAJA3 (Supplementary Fig. 5e).

To further test the chaperone activity of LONP1 and mtHSP70, we chose CLPX as an additional, potential substrate. CLPX was identified from proteomic analysis of mitochondrial protein aggregates in LONP1 knockdown cells (Supplementary Fig. 2e) and verified by Western blot analysis (Supplementary Fig. 5f). CLPX was expressed in vitro, in the presence of LONP1 S855A and/or the mtHSP70 chaperone components (Fig. 5e, f). When present individually, either LONP1 S855A or the mtHSP70 chaperone complex was able to solubilize a fraction of CLPX. The ATPase activity of LONP1 was required for CLPX solubilization (Supplementary Fig. 5g), and LONP1 S855A interacted with solubilized CLPX (Supplementary Fig. 5h). Unlike the case with OXA1L, however, we did not observe synergist activity when LONP1 and mtHSP70 were added to the same reaction. Taken together, these results suggest that LONP1 has an intrinsic chaperone activity on OXA1L and CLPX, and that its ability to synergize with mtHsp70 is substrate-specific.

## Discussion

Our studies suggest that LONP1 and mtHSP70 likely work together to facilitate folding of a substantial fraction of the matrix and inner membrane proteome. In the absence of this system, our mass spectrometry analysis indicates that about 10% of the mitochondrial proteome is substantially aggregated. LONP1 alone, but not its ATPase mutant, has detectable ability to solubilize OXA1L and CLPX in our in vitro assay, suggesting an intrinsic chaperone activity that is dependent on its AAA+ ATPase function. The in vitro assay additionally suggests that LONP1 collaborates with mtHSP70 in distinct ways depending on the substrate. With OXA1L, we detected synergistic protein solubilization when LONP1 and mtHSP70 were present in the same reaction. With CLPX, each component individually had chaperone-like activity, but we were unable to detect synergism. In cells, we find that LONP1 is necessary for the stability of mtHSP70 and DNAJA3, unless mitochondrial protein import is inhibited. We suggest that, in the absence of LONP1, mtHSP70 engages in futile folding interactions with mitochondrial precursors that lead to co-aggregation with substrates. Therefore, LONP1 promotes the protein folding activity of the mtHSP70 chaperone and appears to protect the system from client-induced co-aggregation (Supplementary Fig. 5i).

One caveat to our in vitro assay is that we are not able to test WT LONP1 for folding ability due to its proteolytic activity. We speculate that the interaction of LONP1 with folding substrates versus proteolytic substrates is normally controlled by regulatory mechanisms absent in our minimal in vitro system[32]. One possibility is that LONP1 activity can be regulated to meet cellular demands. Akt phosphorylates LONP1 under stress conditions to enhance the protease activity of LONP1[33]. In normal growth conditions, LONP1 might function, to a larger degree, in protein folding in collaboration with mtHSP70. When cell growth is inhibited and protein misfolding inside mitochondria is increased due to cellular stress, LONP1 phosphorylation may promote a greater role in the removal of misfolded/damaged proteins.

Our results provide a plausible explanation for the observation that Lon protease in yeast is necessary for the assembly of respiratory chain complexes, which led to the suggestion that Lon protease may act as a chaperone[16]. Indeed, in our mass spectrometry analysis, we found 29 subunits of oxidative phosphorylation complexes I to V that became significantly insoluble upon LONP1 inhibition (Supplementary Data 1). It is recognized that human syndromes caused by pathogenic variants in *LONP1* and *HSPA9*, encoding mtHSP70, share common clinical manifestations[34]. These commonalities likely reflect the collaboration of LONP1 and mtHSP70 in the biogenesis of mitochondrial matrix proteins.

## Methods

**Reagents and antibodies**. Reagents: 2-cyano-3,12-dioxo-oleana-1,9(11)-dien-28-oic acid (CDDO) (Cayman Chemical) and carbonyl cyanide 3-chlorophenylhydrazone (CCCP) (Sigma-Aldrich), cycloheximide (Sigma-Aldrich), oligomycin (Sigma-Aldrich), MKT-077 (Sigma-Aldrich). Antibodies for Western blot analysis: LONP1 (Proteintech, 1:3000), OXA1L (Proteintech 1:10000), mtHSP70 (Proteintech, 1:10000), GRPEL1 (Proteintech, 1:5000), TIM44 (Proteintech, 1:5000), CLPX (Proteintech, 1:3000), VDAC1 (Abcam, 1:3000), NDUFA9 (Abcam, 1:3000), DNAJA3 (Santa Cruz BioTech, 1:1000), HSP60 (Santa Cruz BioTech, 1:3000), TOM20 (Santa Cruz BioTech, 1:3000), anti-FLAG M2 (Sigma-Aldrich, 1:10000), anti-HA.11 (Covance, 1:5000).

**Cell culture**. 143B, HeLa, U2OS and LONP1 patient cells were cultured in Dulbecco's modified Eagle's medium (DMEM) supplemented with 10% fetal bovine serum, 50 μg/ml uridine, 2 mM pyruvate and 2 mM glutamine. For glucose or galactose culture, the base DMEM solution lacked glucose, pyruvate, and glutamine (Invitrogen, A14430) and was supplemented with 10% dialyzed FBS, 2 mM glutamine, and 10 mM glucose or galactose.

**Doxycycline-inducible expression**. The Retro-X™ Tet-On® Advanced system (Clontech) was used to establish 143B cells with doxycycline-inducible expression of LONP1-FLAG, OXA1L-FLAG and mtHSP70-HA. Cells were incubated with doxycycline (0.1 μg/ml for OXA1L-FLAG and mtHSP70-HA; as indicated concentration for LONP1-FLAG) to induce protein expression.

**Triton X-100 extraction**. Cells were detached with trypsin, pelleted, and lysed with Triton X-100 buffer (20 mM Tris-HCl, pH 7.4, 150 mM NaCl, 2 mM EDTA, 1% Triton X-100) containing HALT protease inhibitors (Thermo-Pierce). Cell lysates were centrifuged at 20,817g for 10 min at 4 °C to separate into supernatant and pellet fractions. The supernatant and insoluble pellet were reconstituted in equivalent volumes of sample buffer and analyzed by SDS-PAGE and immunoblotting.

**Plasmid constructs and mutagenesis**. OXA1L-FLAG, LONP1-FLAG and mtHSP70-HA were expressed from the mammalian vectors pQCXIP (Clontech) or pRetro-Tight-Pur (Clontech). Vectors were digested with NotI and AgeI (New England Biolabs) and ligated with the following fragments: the LONP1 gene, PCR-amplified from the pcDNA-LONP1 vector (kindly provided by Carolyn Suzuki, Rutgers New Jersey Medical School); the OXA1L or mtHSP70 genes, PCR-amplified from a human cDNA library. PCR primers were designed to add a FLAG or HA tag at the C-terminus. Point mutations were introduced into the coding sequence of LONP1 with overlap extension PCR-based mutagenesis.

For in vitro expression of OXA1L and CLPX, the DHFR control vector included in the PURExpress kit (New England Biolabs) was digested with NdeI and NotI or BamHI (New England Biolabs), and ligated with the OXA1L (amino acid residues 72–435) gene and CLPX (amino acid residues 57–633) gene, PCR-amplified from a human cDNA library.

For bacterial expression of DNAJA3, pETDUET-1 (Novagen) was digested with BamHI and NotI (New England Biolabs), and ligated with the short isoform of the DNAJA3 (amino acid residues 66–453) gene, PCR-amplified from a human cDNA library. All primers used for the cloning and LONP1 mutagenesis are listed in Supplementary Table 1.

**Recombinant protein expression and purification**. Recombinant proteins were expressed in Rosetta (DE3) BL21 *Escherichia coli* (EMD Millipore).

For His-tagged mtHSP70 expression, pET28a-mtHSP70 and pET23a-HEP1 (kindly provided by Júlio César Borges, University of São Paulo) were co-transformed into bacteria[1]. For His-tagged GRPEL1, LONP1 and DNAJA3 expression, pET-His6-GRPEL1 (kindly provided by Henna Tyynismaa, University of Helsinki), pPROEX-LONP1 (kindly provided by Carolyn Suzuki, Rutgers New Jersey Medical School) and pETDUET-1-DNAJA3 vectors were transformed into bacteria. Transformed bacteria were grown in 50 ml terrific broth containing 25 μg/ml chloramphenicol and either 100 μg/ml ampicillin or 50 μg/ml kanamycin at 37 °C to an $OD_{600}$ of 1.2, shifted to 16 °C (for LONP1, mtHSP70 and DNAJA3) or 30 °C (for GRPEL1), and induced with 0.5 mM isopropyl β-d-1-thiogalactopyranoside for 18 h. Cells were harvested by centrifugation at 3500 rpm for 30 min with a swinging bucket rotor (BioFLex HC, Thermo Fisher), and pellets were stored at −20 °C. For purification, bacterial pellets were thawed and resuspended in buffer (25 mM Tris-HCl, pH 7.5, 200 mM NaCl, 10% glycerol, and HALT protease inhibitors (Thermo-Pierce)), lysed by sonication, and cleared by centrifugation at 50,000g for 30 min at 4 °C in Oakridge tubes in the A27-8 × 50 rotor (Thermo Fisher). The soluble supernatant was incubated with washed TALON beads (Clontech) for 4 h at 4 °C. After extensive washing of the beads with buffer (25 mM Tris-HCl, pH 7.5, 200 mM NaCl, 10% glycerol, 20 mM imidazole), proteins were eluted from TALON beads with elution buffer (25 mM Tris-HCl, pH 7.5, 200 mM NaCl, 10% glycerol, 250 mM imidazole). The eluted proteins were further concentrated in buffer (25 mM Tris-HCl, pH 7.5, 200 mM NaCl, 10% glycerol, 25 mM imidazole) by Amicon Ultra Centrifugal Filters (Millipore). Proteins were flash frozen and stored at −80 °C.

**Isolation of mitochondria**. Mitochondria were isolated from cells by differential centrifugation. Cells were collected by scraping in isolation buffer (220 mM mannitol, 70 mM sucrose, 80 mM KCl, 1 mM MgCl₂, 0.5 mM EDTA, 10 mM K + HEPES, pH7.4, and HALT protease inhibitors) and Dounce homogenized (25 strokes) on ice. Cell debris and nuclei were removed by centrifugation twice at 650g for 5 min at 4 °C. A crude mitochondrial fraction was pelleted by centrifugation at 10,000g for 10 min at 4 °C and washed once in isolation buffer.

**RNA interference**. For shRNA-mediated stable knockdown of LONP1 or DNAJA3, the indicated cell lines were infected with retrovirus expressing shRNA from the human H1 promoter. The targeted sequences were: LONP1-Sh (5′-GGG ACATCATTGCCTTGAACC-3′), DNAJA3-sh (5′-GCTGTTCAGGAAGATCTT TGG-3′), CLPP- sh (5′-GCTCAAGAAGCAGCTCTATAA-3′), CLPX- sh (5′-GG AGTATGACTCTGGAGTTGA-3′), non-target shRNA (5′-CGTTAATCGCGTAT AATACGC-3′). For LONP1 small interfering RNA (siRNA) experiments, dicer-substrate RNA (DsiRNA) duplexes (5′-ATCATTGCCTTGAACCCTCTCTAC A-3′) were purchased from Integrated DNA Technologies (IDT). The following predesigned DsiRNAs were purchased from IDT: negative control DsiRNA, hs.Ri.

DNAJA3.13.2, hs.Ri.HSPA9.13.2, hs.Ri.HSPD1.13.1, hs.Ri.GRPEL1.13.3 and hs.Ri.TIMM44.13.2. Transfections of siRNA were performed using Lipofectamine RNAiMAX reagent (Invitrogen) according to the manufacturer's protocol.

**In vitro translation of OXA1L and CLPX**. In vitro translation of OXA1L and CLPX was performed using the PURExpress® In Vitro Protein Synthesis Kit (New England Biolabs). Reactions for OXA1L and CLPX expression were incubated in the absence or presence of recombinant LONP1 and mtHSP70 chaperone components for 8 h at 37 °C and ultracentrifuged at 186,000g for 1 h at 4 °C to separate into supernatant and pellet fractions.

**Immunostaining and Imaging**. Cells were fixed in 10% formaldehyde, permeabilized in 0.1% Triton X-100, and immunostained with antibody against FLAG M2 (Sigma, 1:500), HA.11 (Covance, 1:500), TOM20 (Santa Cruz BioTech, 1:500) or HSP60 (Santa Cruz BioTech, 1:500). Immunofluorescent images were obtained with a Zeiss LSM 710 confocal microscope (Carl Zeiss). Image data were analyzed using ImageJ (v. 1.51J8).

**Intracellular ATP measurement**. The ATP Determination Kit (Thermo Fisher Scientific) was used for detection of intracellular ATP in cell lysates. Cells were resuspended in 0.1 M Tris-HCl, pH 7.4 and lysed by sonication with three cycles of 10 sec on and 30 s off at 20% amplitude (MISONIX S-4000 sonicator). Cell lysates were used for intracellular ATP measurement. Values were normalized to protein concentrations of the lysates. Results were collected from three independent cultures for each sample. Data are presented as means ± standard deviation.

**Co-immunoprecipitation**. Trypsinized cells expressing FLAG-tagged mtHSP70 or LONP1 were lysed in immunoprecipitation (IP) buffer (20 mM Tris-HCl, pH 7.4, 150 mM NaCl, 1 mM EDTA, 5% glycerol, 1% Igepal CA-630) plus HALT protease inhibitors (Thermo-Pierce). The cell lysate was cleared by centrifugation at 20,817g for 10 min at 4 °C, and the supernatant was mixed with ANTI-FLAG® M2 Affinity Gel (Sigma-Aldrich) for 30 min at 4 °C. For immunoprecipitation of in vitro translated OXA1L and CLPX, the reaction was cleared by ultracentrifugation at 186,000g for 1 h at 4 °C and the supernatant was incubated with anti-OXA1L antibody (Proteintech) or anti-LONP1 antibody (Proteintech) for 2 h at 4 °C. Protein A/G-agarose (Thermo-Pierce) was subsequently added and incubated for additional 1 h at 4 °C to capture the immuno-complexes. After the beads were washed three times with IP buffer, bound proteins were eluted by boiling in SDS-PAGE sample buffer and analyzed by SDS-PAGE and immunoblotting. Input samples were collected prior to mixing with beads and 10% (for cell lysates) or 20% (for in vitro translation reactions) of the input samples were used for immunoblot analysis.

**Mass spectrometry analyses of mitochondrial protein aggregates**. Mitochondria were isolated in biological triplicates from control, LONP1 or DNAJA3 knockdown, and CDDO ± cycloheximide (CHX)-treated 143B cells and lysed using Triton X-100 extraction. Insoluble mitochondrial protein pellets were separated by centrifugation at 20,817g for 10 min at 4 °C and solubilized in urea buffer (8 M urea, 100 mM Tris-HCl, pH 8.5). The solubilized samples were digested with trypsin overnight at room temperature and desalted by HPLC with a Microm Bioresources C8 peptide Microtrap. Subsequently, the samples were subjected to LC-MS/MS analysis on a nanoflow LC system, EASY-nLC 1200, (Thermo Fisher Scientific) coupled to a Q Exactive HF Orbitrap mass spectrometer (Thermo Fisher Scientific, Bremen, Germany) equipped with a Nanospray Flex ion source. Samples were directly loaded onto a PicoFrit column (New Objective, Woburn, MA) packed in-house with ReproSil-Pur C18AQ 1.9 μm resin (120 Å pore size, Dr. Maisch, Ammerbuch, Germany) heated to 60 °C or an Aurora 25 cm × 75 μm ID, 1.6 μm C18 column (Ion Opticks, Victoria, Australia) heated to 50 °C. The peptides were separated with a 60 min gradient at a flow rate of 220 nL/min for the in-house packed column or 350 nL/min for the Aurora column. The gradient was as follows: 2–6% Solvent B (3.5 min), 6–25% B (41.5 min), and 25–40% B (15 min) and to 100% B (10 min). Solvent A consisted of 97.8% $H_2O$, 2% ACN, and 0.2% formic acid and solvent B consisted of 19.8% $H_2O$, 80% ACN, and 0.2% formic acid. The Q Exactive HF Orbitrap was operated in data dependent mode with the Tune (version 2.7 SP1build 2659) instrument control software. Spray voltage was set to 2.5 kV, S-lens RF level at 50, and heated capillary at 275 °C. Full scan resolution was set to 60,000 at m/z 200. Full scan target was $3 × 10^6$ with a maximum injection time of 15 ms. Mass range was set to 400–1650 m/z. For data dependent MS2 scans collected at 30,000 resolution the loop count was 12, target value was set at $1 × 10^5$, and intensity threshold was kept at $1 × 10^5$. Isolation width was set at 1.2 m/z and a fixed first mass of 100 was used. Normalized collision energy was set at 28. Peptide match was set to off, and isotope exclusion was on. Data acquisition was controlled by Xcalibur (4.0.27.13) and all data were acquired in profile mode.

**Data analysis**. Raw data were searched in Proteome Discoverer 2.4 (Thermo Scientific) using the Byonic search algorithm (Protein Metrics) and Uniprot human database. PD-Byonic search parameters were as follows: fully Tryptic peptides with no more than 2 missed cleavages, precursor mass tolerance of 10 ppm and

fragment mass tolerance of 20 ppm, and a maximum of 2 common modifications and 2 rare modifications. Cysteine carbamidomethylation was set as a static modification, while methionine oxidation was a common dynamic modification (up to 2 per peptide). Methionine loss on protein N-termini, methionine loss + acetylation on protein N-termini, protein N-terminal acetylation, lysine acetylation, and phosphorylation of serine, threonine, and tyrosine were rare dynamic modifications (only 1 each). Byonic protein-level FDR was set at 0.01, while Percolator FDRs were set at 0.01 (strict) and 0.05 (relaxed). In the consensus step, peptide and PSM FDRs were set at 0.001 (strict) and 0.01 (relaxed), with peptide confidence at least medium, lower confidence peptides excluded, minimum peptide length set at 6, and apply strict parsimony set to false.

LFQ was then performed with the Minora feature detector, feature mapper, and precursor ions quantifier nodes. Retention time alignment was performed with maximum RT shift of 5 min and a minimum S/N threshold of 10. Quantified peptides included unique + razor, protein groups were considered for peptide uniqueness, shared Quan results were not used, Quan results with missing values were not rejected, and precursor abundance was based on intensity. Abundances were normalized using the total intensities of all non-mitochondrial proteins identified in a pre-search of the raw data files from the control, LONP1 kd, DNAJA3 kd and CDDO ± CHX samples with identical search parameters. Imputation was then performed using the low abundance resampling method.

Statistical analysis of protein abundances was performed using methods developed for linear models for microarray data[35] available as a package in R (R version 4.0.2, limma version 3.44.3). Samples were normalized between arrays, modeled for expression using the linear model fit, and fold changes were calculated as a log2 function, and multiple testing correction utilized the Benjamini-Hochberg method to calculate FDR-adjusted p-values[36]. Proteins and peptides identified are listed in Supplementary Data 3 and 4. Proteins were then annotated as being localized to the mitochondria based on annotation in either MitoCarta or UniProt, and only those mitochondrial proteins with 2 or more unique peptides and quantified in at least 5 of the 9 replicates from the LONP1 kd, DNAJA3 kd, and CDDO samples were subsequently retained (Supplementary Data 1) for further analysis and included in figure generation. Proteins considered significant were enriched at least 2-fold over control with a BH-adjusted p-value < 0.05. Mass spectrometry data files have been deposited at ProteomeXchange, with access code PXD021939.

**Statistics and reproducibility**. For Western blotting, confocal microscopy, ATP measurement, and mass spectrometry, all experiments were independently repeated 2 or 3 times.

**Reporting summary**. Further information on research design is available in the Nature Research Reporting Summary linked to this article.

## Data availability

The mass spectrometry proteomics data have been deposited to the ProteomeXchange Consortium via the PRIDE partner repository with the dataset identifier PXD021939. Source data are provided with this paper.

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

## Competing interests
The authors declare no competing interests.

## Additional information

**Peer review information** *Nature Communications* thanks Cheryl F. Lichti, Carolyn Suzuki, and other, anonymous, reviewers for their contributions to hte peer review of this work. Peer review reports are available.

