## [Peer Review File · Nature Communications]

REVIEWER COMMENTS

Reviewer #1 (Remarks to the Author):

Many mitochondrial proteins require the assistance of chaperones to fold into their native conformation after their import into mitochondria. Moreover, proteases remove misfolded and/or aggregated protein species and thereby contribute to protein homeostasis within mitochondria. Shin et al. report here that the AAA+ protease LONP1, in addition to its proteolytic function, plays a direct role in protein folding by cooperating with mitochondrial HSP70. They initially observe that knockdown or chemical inhibition of LONP1 leads to widespread protein aggregation in the mitochondrial matrix, which can be rescued by a proteolytically inactive mutant of LONP1. Among the proteins that become insoluble upon LONP1 knockdown are mtHSP70 and its cochaperones DNAJA3. This and the fact that LONP1 and mtHSP70 physically interact *in vivo* suggests that LONP1 might be required for mtHSP70 solubility and function. In an *in vitro* reconstitution experiment, they demonstrate that the solubility of OXA1L, when translated in a cell-free system, is increased in the presence of LONP1 alone, but in particular when both LONP1 and mtHSP70 are added to the reaction. Shin et al. conclude that LONP1 is crucial for a productive mtHSP70 folding pathway. This study is a 're-discovery' of a well-known phenomenon: Protease mutants without proteolytic activity often exhibit substantial chaperone activity. In the case of LONP1 and other mitochondrial proteases, this was prominently published in the past in technically excellent studies of the Schatz, Langer and Grivell labs (e.g. Rep et al. *Science* 1996; Leonhard et al. *Nature* 1999). For the membrane-embedded AAA enzymes, the i-AAA and m-AAA proteases, there is some evidence for a non-proteolytic activity due to their role in the import of specific precursor proteins. However, for the LONP1 protease, a non-proteolytic activity has – to my knowledge – only been proposed but not been demonstrated so far. Evidence for such a physiological role of LONP1 as chaperone is also not convincingly demonstrated here. The authors also disregard the function of the ClpXP system which comprises foldase and protease activity and functionally interacts with LONP1 as well as with the HSP70 system.

The experiments are of high quality and the data are presented in a clear and concise way. The idea that LONP1 cooperates with mtHSP70 for productive protein folding is certainly of interest for a general readership. However, the mechanistic insights into the chaperone-like function of LONP1 are somewhat superficial and the *in vivo* relevance of the protein folding activity of wildtype LONP1 (in comparison to its proteolytic activity) is not really clarified by the presented work. The main conclusion of this study does not go beyond those from previous publications and thus, this study is of good quality but mainly confirmative.

Major points:

1

It has been a longstanding observation that proteases can act as chaperones when their proteolytic function is abolished by mutations, but the relevance of these observations remained unclear. This manuscript therefore addresses an important and interesting question and sheds new light on the role of proteases in mitochondrial protein folding. Whereas many of the data shown are similar to those in previous studies, the proteomic analysis of potential LONP clients is novel (as proteomics was not possible in the 90s). The authors show using OXA1L and HSP70 as model substrates that they can distinguish between aggregation-prone and aggregate-binding proteins. They should use these conditions for a proteomic analysis to further group the proteins in the aggregates.

2

Along the same line, they show that they can distinguish LONP substrates that are newly imported from those that are aggregation-prone even after being folded. Again, proteomics could be very interesting to separate these groups of proteins. A number of aggregation-prone mitochondrial proteins were described in the past, however, their potential interaction with LONP was never studied.

3

For the proteomic studies, at least three replicates are required and volcano plots should be shown. Examples of particular aggregation-prone and/or LONP-dependent proteins should be specifically mentioned here since this will be of interest for further studies. In the current manuscript this information is only available from the supplement. Moreover, the entire excel list of proteomic data should be presented, not only a selected hit list of mitochondrial candidates.

4

The proteolytic inactive mutant was highly overexpressed in this study. It needs to be demonstrated that the (partial) suppression is also observed at endogenous levels. This needs to be properly controlled and addressed by additional experimental evidence.

Minor points:

5

Can it be excluded that the LONP1S855A mutant has some residual protease activity? Since the authors overexpress it here, even a small residual proteolytic activity of this mutant could be sufficient to restore degradation of misfolded or aggregated proteins. This is essential for all mechanistic conclusions.

6

The in vitro reconstitution experiment shown in Figure 4 is nice, but raises some questions:

- First, why are the samples centrifuged for 1 hour at 186,000g to pellet aggregates? Is the pellet fraction really aggregates, couldn't this also be ribosome-associated proteins?
- Second, what does "without OXA1L" in 4c, leftmost lane mean? Is the in vitro translation system present, but without mRNA? An even better control would be to add an mRNA coding for an "inert" protein that neither aggregates in this assay nor requires mtHSP70 or LONP1 for folding (e.g. a soluble cytosolic protein).
- Third, upon addition of LONP1S588A, mtHSP70 and OXA1L not only change in solubility, but also considerably in total abundance. Why is that? Reduced translation of OXA1L (which could affect its aggregation propensity!)? Residual proteolytic activity of the LONP1 mutant?
- Fourth, as the authors state, OXA1L can never fold properly in this setting due to the lack of a membrane. Wouldn't this experiment be more informative with a soluble substrate, for which they identified many potential candidates in their mass spectrometry experiment from Figure 2b?

7

How does LONP1 "decide" which substrates to (un)fold and which to degrade? The authors only speculate very generically that there might be some regulatory mechanism. One or several more specific hypotheses would be nice that can be further investigated in future studies.

8

The authors identified a number of intermembrane space proteins in the aggregates. They also listed a number of these proteins as proteins of unknown location or matrix which is incorrect (for example CHCHD2, CHCHD3). While these proteins might be artificially trapped here as post-lysis binding artifacts, this observation could also indicate some cross-talk between the different mitochondrial sub-compartments. This should be discussed.

Reviewer #2 (Remarks to the Author):

The manuscript submitted by Shin et al., entitled "LONP1 and mtHSP70 cooperate to promote mitochondrial protein folding" addresses an interesting and important physiological and mechanistic problem as how the ATP-dependent LONP1 protease potentially mediates protein folding and assembly within mitochondria. Although other studies have proposed a chaperone-like function of LONP1, no subsequent work has rigorously demonstrated LONP1's chaperone activity. The current work argues

that LONP1 is required for the solubility and function of the mtHSP70, DNAJA3 and GRPEL1 protein folding pathway. To support this proposition, cell culture experiments were performed using siRNA knockdown of LONP1, DNAJA3 and GRPEL1, along with rescue experiments overexpressing wild type LONP1 and LONP1 mutants that have been previously reported to lack protease or ATPase activity, and also treatment of cells with pharmacologic inhibitors of LONP1. In addition, reconstitution experiments using recombinant proteins isolated from bacteria were performed to examine protein solubility and aggregation.

Convincing data are presented that LONP1 interacts with the mtHSP70 system in 143B osteosarcoma cells. In its current form, however, the study does not provide strong evidence demonstrating that LONP1 mediates the folding/assembly of the mtHSP70 machinery and/or OXA1L1 and NDUFA9- two proteins showing increased insolubility upon LONP1 knockdown. The siRNA knockdown of LONP1 leads to a striking insolubility of both DNAJA3 long and short isoforms (Fig. 2a and d). Insolubility of mtHSP70 is less dramatic (Fig. 2d), and the effect of LONP1 knockdown on GRPEL1 was not shown. The knockdown of GRPEL1 shows increased co-immunoprecipitation with mtHSP70-flag with LONP1 and DNAJA3 (Fig. 3a). This is likely because down-regulation of GRPEL1 mediated ADP exchange by mtHSP70 stabilizes interactions with this protein. Whether a chaperone-like activity of LONP1 is important for solubility of mtHSP70 and DNAJA3 is unclear based on the data in Fig. 3 d and e. An important control is missing from these LONP1 siRNA experiments, which is to overexpress not only the WT and S855A protease mutant as shown, but also to overexpress the K529R ATPase mutant. As the latter LONP1 mutant is predicted to have impaired ATP-dependent chaperone activity, this control would strengthen the interpretation presented by the authors. It is interesting that the knockdown of DNAJA3 leads to insolubility of OXA1L, NDUFA9 and other mitochondrial proteins (determined by mass spec), which is similar to that observed when LONP1 is knocked down (Fig. 2b and c). However, the observed increased protein insolubility of the LONP1 knockdown maybe attributed to the primary dysfunction of the mtHSP70 system and/or the absence of LONP1 protein quality control, rather than defects in a chaperone-like function of LONP1. Again, expressing the LONP1 K529R ATPase mutant in the LONP1 knockdown cells should be examined.

Although, in vitro reconstitution is a solid approach to demonstrate the chaperone-like function of LONP1, the results are not convincing and the experiment requires more controls with individual mtHSP70 proteins (e.g. 70, JA3, EL1). In Fig. 4b, why is LONP1 protein in the pellet and not fully soluble? One wonders if unfolded subunits and/or misassembly of the LONP1 complex aggregate in the pellet fraction. In Fig. 4c, why is there less total OXAL1 protein with 1 uM and 2 uM S855A protease LONP1 mutant? In Fig. 4c, why is there less DNAJA3 short form in the samples with 1 uM and 2 uM LONP1 protease mutant S855A?

Line 200. The following statement requires further experimental support- "Addition of WT LONP1 resulted in degradation of OXA1L, likely because it is recognized as a misfolded substrate (Extended Data Fig. 4a)." To demonstrate that OXAL1 is degraded by LONP1, reactions should be incubated +/- ATP, show that loss of OXAL1 is ATP-dependent. Again, in Extended Data Fig. 4b, one wonders why there is LONP1 in the pellet fraction, and why there is more LONP1 in the pellet when CDDO is present. There is also more OXAL1 in the pellet fraction + CDDO. Therefore a control lane should be included showing the input proteins- S855A, mtHSP70, JA3 and EL1 before and after 1 hour at 37 degrees C.

There are two significant caveats in this study. One caveat is that CDDO is not a LonP1-specific inhibitor (Figs. 1 and 2). The second is that doxycycline (DOX) has been shown to inhibit mitochondrial protein synthesis. Here, DOX is used to induce OXA1L-flag expression, in cells which may confound the interpretation of results. These limitations should be addressed, or alternatives approaches employed.

CDDO has multiple and diverse targets throughout the cell and also within mitochondria. For example, I-kappaB kinase, Jak1 and STAT3 are direct targets of CDDO (see references 1 and 2 at the end of this critique), in addition to other targets (references 3 and 4). In addition, the mechanism by which CDDO inhibits LonP1 has not been published, and it is likely that CDDO blocks other mitochondrial

proteins. Therefore, results from experiments using CDDO are not clearly interpretable. Doxycycline (DOX) has been shown to inhibit mitochondrial protein synthesis in cultured cells at a concentration of 1.0 microgram/ml, which is used in this study (reference 5). This effect of DOX is particularly relevant as OXA1L interacts with mitochondrial ribosomes, mediating the insertion of mitochondrial- as well as nuclear- encoded proteins into the mitochondrial inner membrane. Therefore, in experiments using DOX, an important control would be to determine the protein levels of mtDNA-encoded proteins. Control experiments are also crucial for determining the separate and combined effects of CDDO and DOX on the synthesis and stability of mtHSP70 and DNAJA3 as well as LonP1, OXA1L and NDUFA9.

Minor comments:

Introduction

The Introduction is quite abbreviated. For the benefit of general readers, a brief description of the AAA+ LONP1 protease and its ATPase and proteases activities would be helpful. Also include a brief description of mtHSP70- an ATP-dependent chaperone, its co-chaperone DNAJA3, which stimulates the ATPase activity of mtHSP70 thereby releasing bound protein substrate, and the exchange factor GRPEL1, which stimulates the release hydrolyzes ADP allowing re-binding of ATP. Help for the readers to also briefly describing the localization and function OXA1L and NDUFA9 (e.g. carboxyl-terminus of this OXA1L interacts with mitochondrial ribosomes mediating insertion of both mitochondrial and nuclear encoded proteins into the mitochondrial inner membrane; and NDUFA9 subunit is at the membrane interface of Complex I NADH dehydrogenase, which binds NADH and required for complex assembly or stability).

Line 45. LONP1 has long been "suspected". Perhaps "proposed" is a more accurate word than "suspected" as acting as a chaperone in the assembly of oxidative phosphorylation (OXPHOS) complexes in yeast and mammalian mitochondria. Suggest the following edit- "LONP1 has been proposed to have a chaperone-like function in the assembly of oxidative phosphorylation (OXPHOS) complexes in yeast and mammalian mitochondria, which is independent of its protease activity."

Results

Provide a line about why the study concentrates on OXA1L and NDUFA9.

Figure 1b. Although overexpressing WT and protease mutant LONP1 reduces the insoluble fraction of OXA1L and NDUFA9 considerably, they do not increase the soluble fraction compared to vector. Why might this be observed?

Throughout the results, a good practice would be to include quantification of immunoblots and number of replicates performed.

Line 91 is unclear "Because LONP1 is required for proper OXA1L biogenesis, it is possible that LONP1 inhibition disrupts the function of chaperone systems in the mitochondrial matrix. We therefore tested the status of OXA1L upon disruption of HSP60 and mtHSP70".

Figure 2d- In LONP1 knockdown cells, overexpression of the LONP1 protease mutant S855A leads to decrease in total levels of DNAJA3, although the ratio of pellet: insoluble is less than vector control. Some comments about this would be helpful to the reader.

References

1. J Biol Chem. 28:35764-9 (2006)
Triterpenoid CDDO-Me blocks the NF-kappaB pathway by direct inhibition of IKKbeta on Cys-179.
Ahmad R, Raina D, Meyer C, Kharbanda S, Kufe D.

2. Cancer Res. 68:2920-6 (2008)

Triterpenoid CDDO-methyl ester inhibits the Janus-activated kinase-1 (JAK1)signal transducer and activator of transcription-3 (STAT3) pathway by direct inhibition of JAK1 and STAT3. Ahmad R, Raina D, Meyer C, Kufe D.

3. Pharmacol Rev. 64:972-1003 (2012)

Synthetic oleanane triterpenoids: multifunctional drugs with a broad range of applications for prevention and treatment of chronic disease.

Liby KT, Sporn MB.

4. Nat Rev Cancer. 7:357-69 (2007)

Triterpenoids and rexinoids as multifunctional agents for the prevention and treatment of cancer.

Liby KT, Yore MM, Sporn MB.

5. Cell Rep.10:1681-1691 (2015)

Tetracyclines Disturb Mitochondrial Function across Eukaryotic Models: A Call for Caution in Biomedical Research. Moullan N, Mouchiroud L, Wang X, Ryu D, Williams EG, Mottis A, Jovaisaite V, Frochaux MV, Quiros PM, Deplancke B, Houtkooper RH, Auwerx J.

Reviewer #3 (Remarks to the Author):

This manuscript by Shin et al concerns the Lon protease of human mitochondria, LONP1. Specifically, the authors probe a role of Lon in promoting protein folding, making use of a variant lacking proteolytic activity. Overall the data presented in the manuscript makes a good case for interplay between LONP1 and the mitochondrial Hsp70 system in a prevention of aggregation/protein folding pathway. Thus, this manuscript could make an important contribution to the field, because although the idea that Lon has "chaperone-like" activity has been discussed, there has been little solid evidence to support it. However, there is a serious issue with the writing that leads to much confusion about how the authors are interpreting their data and what they are trying to say. In several cases, particularly in the "results" sections, terms (for example, "physical interaction"; "aggregation") are used in a way that causes these problems. A serious rewriting is imperative.

Specific comments regarding experimental data:

1. Why is DNAJA3 not identified in the mass spec data, when it is so obvious in the western blots shown in fig 2a?

2. Related to comment 1, and as a more general comment for several experiments. It is important to show total as well as sup and pellet fractions as a control to confirm that close to 100% of the starting material is being recovered.

3. As a control, analysis of the effect of depletion of ClpX would be valuable. Presumably, the Hsp70 and DNAJA3 would not be present in aggregates and thus bolster the authors' conclusion that LONP1 is acting specifically in the Hsp70 pathway.

4. Since Hsp70 systems, as well as Lon, are dependent on ATP binding and hydrolysis, in vitro OXA1L solubility assay (Fig.4C in the presence of LONP1 and chaperones) should be done in the presence of excess of either ATP or ADP. The results should be more informative regarding the claims that the ADP state of Hsp70 matters and the ATPase of LONP1 also matters. Note: It is not clear in the M&M or figure legend what nucleotide was present in excess.

Specific major comments on writing.

5. A major cause of confusion in this manuscript is the use of the term "Hsp70 solubility" (as in Line 122, the heading "Lon is required for mtHsp70 solubility"; also lines 128-132) for the case when Hsp70 is found in protein aggregates. For the first two reads through the results section this reviewer thought that the authors were trying to make the case that Lon was necessary to keep Hsp70 properly

folded and active itself (that is was a chaperone for Hsp70). Using the same term for the case when a protein substrate of the chaperone system aggregates and when a chaperone is actively involved in trying to resolubilize aggregates/prevent aggregation should be avoided.

6. Another cause of confusion is that the authors consider that anything that comes down in a pull-down is "physically interacting". An example is Line 153, heading "Lon physically interacts with mtHsp70". Because finding actual physical interactions (that is the direct physical interaction of one protein with another) between chaperone systems is a very active field of research, using this term of things coming down together in large amorphous aggregates is very confusing.

7. These and other problems could be helped by a suitable introduction to the Hsp70 system – how it works, what happens upon substrate aggregation, etc. A standard introduction would set the framework for how terms are being used in the results sections.

Minor points.

8. Line 111 -According to the source file with MS results OXA1L was 3.380921505- and 7.162547035 - fold increased - not 3.9.

9. Line 114 – not clear where the statement that correlation coefficient equals 0.902 comes from; in Fig.2c on the graph it is $R^2 = 0.8139$.

10. Line 125 - "Supplementary" Table instead of "Extended Data"

11. Line 130/132 -This sentence comes out of nowhere. The experimental approach, including use of previously unmentioned inhibitor, should be explained more.

REVIEWER COMMENT

Reviewer #1 (Remarks to the Authors):

mitochondrial proteins require the assistance of chaperones to fold into their native conformation after their import into mitochondria. Moreover, proteases remove misfolded and/or aggregated protein species and thereby contribute to protein homeostasis within mitochondria. Shin et al. report here that the AAA+ protease LONP1, in addition to its proteolytic function, plays a direct role in protein folding by cooperating with mitochondrial HSP70. They initially observe that knockdown or chemical inhibition of LONP1 leads to widespread protein aggregation in the mitochondrial matrix, which can be rescued by a proteolytically inactive mutant of LONP1. Among the proteins that become insoluble upon LONP1 knockdown are mtHSP70 and its cochaperones DNAJA3. This and the fact that LONP1 and mtHSP70 physically interact *in vivo* suggests that LONP1 might be required for mtHSP70 solubility and function. In an *in vitro* reconstitution experiment, they demonstrate that the solubility of OXA1L, when translated in a cell-free system, is increased in the presence of LONP1 alone, but in particular when both LONP1 and mtHSP70 are added to the reaction. Shin et al. conclude that LONP1 is crucial for a productive mtHSP70 folding pathway.

This study is a 're-discovery' of a well-known phenomenon: Protease mutants without proteolytic activity often exhibit substantial chaperone activity. In the case of LONP1 and other mitochondrial proteases, this was prominently published in the past in technically excellent studies of the Schatz, Langer and Grivell labs (e.g. Rep et al. *Science* 1996; Leonhard et al. *Nature* 1999). For the membrane-embedded AAA enzymes, the i-AAA and m-AAA proteases, there is some evidence for a non-proteolytic activity due to their role in the import of specific precursor proteins. However, for the LONP1 protease, a non-proteolytic activity has – to my knowledge – only be proposed but not been demonstrated so far. Evidence for such a physiological role of LONP1 as chaperone is also not convincingly demonstrated here. The authors also disregard the function of the ClpXP system which comprises foldase and protease activity and functionally interacts with LONP1 as well as with the HSP70 system.

The experiments are of high quality and the data are presented in a clear and concise way. The idea that LONP1 cooperates with mtHSP70 for productive protein folding is certainly of interest for a general readership. However, the mechanistic insights into the chaperone-like function of LONP1 are somewhat superficial and the *in vivo* relevance of the protein folding activity of wildtype LONP1 (in comparison to its proteolytic activity) is not really clarified by the presented work. The main conclusion of this study does not go beyond those from previous publications and thus, this study is of good quality but mainly confirmative.

Major points:

1

It has been a longstanding observation that proteases can act as chaperones when their proteolytic function is abolished by mutations, but the relevance of these observations remained unclear. This manuscript therefore addresses an important and interesting question and sheds new light on the role of proteases in mitochondrial protein folding. Whereas many of the data shown are similar to those in previous studies, the proteomic analysis of potential LONP clients is novel (as proteomics was not possible in the 90s). The authors show using OXA1L and

HSP70 as model substrates that they can distinguish between aggregation-prone and aggregate-binding proteins. They should use these conditions for a proteomic analysis to further group the proteins in the aggregates.

2

Along the same line, they show that they can distinguish LONP substrates that are newly imported from those that are aggregation-prone even after being folded. Again, proteomics could be very interesting to separate these groups of proteins. A number of aggregation-prone mitochondrial proteins were described in the past, however, their potential interaction with LONP was never studied.

Response to 1 and 2: It is a good suggestion to use proteomics to distinguish between proteins that require LONP1 function during their biogenesis versus those that are aggregation-prone even after folding. Using mass spectrometry, we compared mitochondrial aggregates from cells treated with CDDO versus cells treated with CDDO and cycloheximide. CDDO treatment caused an aggregation profile highly similar to LONP1 knockdown (Fig. 2b, d). Interestingly, co-treatment with cycloheximide prevented most CDDO-induced aggregation, suggesting that the majority of CDDO-induced aggregation occurs during the biogenesis of imported mitochondrial proteins. Out of 186 proteins enriched in mitochondrial aggregates in CDDO-treated cells, only 11 mitochondrial proteins remained aggregated with CHX co-treatment (Fig. 2d, Supplementary Fig. 2e).

3

For the proteomic studies, at least three replicates are required and volcano plots should be shown. Examples of particular aggregation-prone and/or LONP-dependent proteins should be specifically mentioned here since this will be of interest for further studies. In the current manuscript this information is only available from the supplement. Moreover, the entire excel list of proteomic data should be presented, not only a selected hit list of mitochondrial candidates.

Response: We repeated the mass spectrometric analysis of mitochondrial protein aggregates, and now the proteomic data of Fig. 2 and Supplementary Fig. 2 are from triplicate datasets. Volcano plots are shown in Supplementary Fig. 2e. In addition, we provide our entire mass spectrometric data containing 455 mitochondrial proteins in Supplementary Table 1. We discuss and point out in Supplementary Fig. 2e that UQCRC1, CLPX and MRPL23 were found in common in LONP1 knockdown, DNAJA3 knockdown and CDDO-treated cells. However, co-treatment with cycloheximide prevented their aggregation (Supplementary Fig. 2e). These results suggest that LONP1 and the mtHSP70 chaperone are required for the folding process of UQCRC1, CLPX and MRPL23 during their biogenesis phase. We go on to analyze CLPX in more detail, as described below in the response to point 6.

4

The proteolytic inactive mutant was highly overexpressed in this study. It needs to be demonstrated that the (partial) suppression is also observed at endogenous levels. This needs to be properly controlled and addressed by additional experimental evidence.

Response: To address this issue, we generated a DOX-inducible system for LONP1 expression and titrated DOX concentration to control the expression level of siRNA-resistant LONP1 (WT, S885A and K529R). At 0.05 µg/ml of DOX, LONP1 was induced at an endogenous level and rescued the aggregation of OXA1L, NDUFA9, mtHSP70 and DNAJA3 (Fig 1b and 3a). The

protease mutant (S885A) also showed substantial rescue at this level. We confirm that 0.1 $\mu\text{g/ml}$ DOX has no effect on the mtDNA translation product CO1 (Supplementary Fig. 1c).

Minor points:

5

Can it be excluded that the LONP1S855A mutant has some residual protease activity? Since the authors overexpress it here, even a small residual proteolytic activity of this mutant could be sufficient to restore degradation of misfolded or aggregated proteins. This is essential for all mechanistic conclusions.

Response: As shown above, an endogenous level of LONP1 S855A greatly reduced aggregated OXA1L, NDUFA9 and DNAJA3 in LONP1 knockdown cells (Fig 1b and 3a). Importantly, with regard to the current concern, there is also *restoration of these proteins in the soluble fraction*. This latter observation indicates that LONP1 S855A restores solubility of these proteins, versus degrading aggregates.

To address the issue of residual protease activity of LONP1 S855A, we utilized DHFR as a proteolytic substrate. DHFR was expressed via in vitro translation in the presence of purified LONP1 WT or S855A. WT LONP1 at 2 μM significantly reduced the DHFR level, and 8 μM completely degraded DHFR. In contrast, 2 μM LONP1 S855A did not change the level of DHFR. At 8 μM , we detected moderate reduction of the DHFR level, suggesting that LONP1 S855A has low residual protease activity that can only be detected when high concentrations of protein are analyzed. However, as noted above, expression of endogenous levels of LONP1(S885A) results in substantial rescue of substrate solubility, a result that cannot be explained by residual protease activity.

6

The in vitro reconstitution experiment shown in Figure 4 is nice, but raises some questions: - First, why are the samples centrifuged for 1 hour at 186,000g to pellet aggregates? Is the pellet fraction really aggregates, couldn't this also be ribosome-associated proteins?

Response: To address this issue, we compared centrifugation of in vitro translation products at low-speed (20,817 g) versus high-speed (187,000 g). The results for both speeds are similar, indicating that ultracentrifugation (and sedimentation of ribosomes) is not necessary.

- Second, what does “without OXA1L” in 4c, leftmost lane mean? Is the in vitro translation system present, but without mRNA? An even better control would be to add an mRNA coding for an “inert” protein that neither aggregates in this assay nor requires mtHSP70 or LONP1 for folding (e.g. a soluble cytosolic protein).

Response: We agree a better control is to express an inert protein. To address this issue, we expressed DHFR (dihydrofolate reductase) as a control protein in our in vitro assay and tested the solubility of the mtHSP70 chaperone components and LONP1 (Supplementary Fig. 5b).

- Third, upon addition of LONP1S588A, mtHSP70 and OXA1L not only change in solubility, but also considerably in total abundance. Why is that? Reduced translation of OXA1L (which could affect its aggregation propensity!)? Residual proteolytic activity of the LONP1 mutant?

Response: As noted above, we have evidence that the LONP1 S855A mutant retains low residual protease activity. However, as shown in Fig. 5c-f, the change in OXA1L and CLPX abundance is slight.

- Fourth, as the authors state, OXA1L can never fold properly in this setting due to the lack of a membrane. Wouldn't this experiment be more informative with a soluble substrate, for which they identified many potential candidates in their mass spectrometry experiment from Figure 2b?

Response: Yes, it would be ideal to study additional substrates, especially ones that do not require a membrane. To this end, we examined CLPX as a potentially soluble substrate for the chaperone activity of LONP1 and mtHSP70. We identified CLPX from proteomic analysis of mitochondrial protein aggregates (Supplementary Fig. 2e, Supplementary Table 1) and verified its behavior by Western blot analysis (Supplementary Fig. 5f).

We expressed CLPX in vitro, along with LONP1 S855A and/or the mtHSP70 chaperone components (Fig. 5e, f). LONP1 S855A and mtHSP70 were individually able to solubilize CLPX. However, we did not observe synergist solubilization activity when LONP1 and mtHSP70 were added together. These new results indicate that LONP1 and the mtHSP70 chaperone interact with different substrates in distinct ways. We discuss these complexities in the Discussion.

7

How does LONP1 “decide” which substrates to (un)fold and which to degrade? The authors only speculate very generically that there might be some regulatory mechanism. One or several more specific hypotheses would be nice that can be further investigated in future studies.

Response: In the revised Discussion, we bring up some specific possibilities for regulation of LONP1 activity. One possibility is that LONP1 activity can be regulated to meet cellular demands. Akt phosphorylates LONP1 under stress conditions to enhance the protease activity of LONP1¹. In normal growth conditions, LONP1 might function, to a larger degree, in protein folding in collaboration with mtHSP70. When cell growth is inhibited and protein misfolding inside mitochondria is increased due to cellular stress, LONP1 phosphorylation might promote a greater role in the removal of misfolded/damaged proteins.

8

The authors identified a number of intermembrane space proteins in the aggregates. They also listed a number of these proteins as proteins of unknown location or matrix which is incorrect (for example CHCHD2, CHCHD3). While these proteins might be artificially trapped here as post-lysis binding artifacts, this observation could also indicate some cross-talk between the different mitochondrial sub-compartments. This should be discussed.

Response: In our new triplicate datasets, 135 mitochondrial proteins are significantly aggregated both in LONP1 and DNAJA3 knockdown mitochondria (Supplementary Fig. 2c and Table 2). CHCHD3 is now excluded due to a *p*-value >0.05; however, CHCHD2 is still included. We have corrected the subcompartmental localization of CHCHD2 as ‘intermembrane space’, and it is the only protein localized in the intermembrane space (Supplementary Fig. 2d and Table 2).

=====

Reviewer #2 (Remarks to the Author):

The manuscript submitted by Shin et al., entitled “LONP1 and mtHSP70 cooperate to promote mitochondrial protein folding” addresses an interesting and important physiological and mechanistic problem as how the ATP-dependent LONP1 protease potentially mediates protein folding and assembly within mitochondria. Although other studies have proposed a chaperone-like function of LONP1, no subsequent work has rigorously demonstrated LONP1’s chaperone activity. The current work argues that LONP1 is required for the solubility and function of the mtHSP70, DNAJA3 and GRPEL1 protein folding pathway. To support this proposition, cell culture experiments were performed using siRNA knockdown of LONP1, DNAJA3 and GRPEL1, along with rescue experiments overexpressing wild type LONP1 and LONP1 mutants that have been previously reported to lack protease or ATPase activity, and also treatment of

cells with pharmacologic inhibitors of LONP1. In addition, reconstitution experiments using recombinant proteins isolated from bacteria were performed to examine protein solubility and aggregation.

Convincing data are presented that LONP1 interacts with the mtHSP70 system in 143B osteosarcoma cells. In its current form, however, the study does not provide strong evidence demonstrating that LONP1 mediates the folding/assembly of the mtHSP70 machinery and/or OXA1L1 and NDUFA9- two proteins showing increased insolubility upon LONP1 knockdown.

The siRNA knockdown of LONP1 leads to a striking insolubility of both DNAJA3 long and short isoforms (Fig. 2a and d). Insolubility of mtHSP70 is less dramatic (Fig. 2d), and the effect of LONP1 knockdown on GRPEL1 was not shown.

Response: In the revised manuscript we examined the solubility of GRPEL1 in LONP1 knockdown cells and found that GRPEL1 is minimally affected (Supplementary Fig. 3b).

The knockdown of GRPEL1 shows increased co-immunoprecipitation with mtHSP70-flag with LONP1 and DNAJA3 (Fig. 3a). This is likely because down-regulation of GRPEL1 mediated ADP exchange by mtHSP70 stabilizes interactions with this protein. Whether a chaperone-like activity of LONP1 is important for solubility of mtHSP70 and DNAJA3 is unclear based on the data in Fig. 3 d and e.

An important control is missing from these LONP1 siRNA experiments, which is to overexpress not only the WT and S855A protease mutant as shown, but also to overexpress the K529R ATPase mutant. As the latter LONP1 mutant is predicted to have impaired ATP-dependent chaperone activity, this control would strengthen the interpretation presented by the authors.

It is interesting that the knockdown of DNAJA3 leads to insolubility of OXA1L, NDUFA9 and other mitochondrial proteins (determined by mass spec), which is similar to that observed when LONP1 is knocked down (Fig. 2b and c). However, the observed increased protein insolubility of the LONP1 knockdown maybe attributed to the primary dysfunction of the mtHSP70 system and/or the absence of LONP1 protein quality control, rather than defects in a chaperone-like function of LONP1. Again, expressing the LONP1 K529R ATPase mutant in the LONP1 knockdown cells should be examined.

Response: We agree it is important to test whether the ATPase activity of LONP1 is required for rescue of protein solubility. In the revised manuscript, we show that LONP1 K529R, unlike S855A, is incapable of rescuing aggregation of OXA1L, mtHSP70, and DNAJA3 (Fig. 3b).

Although, in vitro reconstitution is a solid approach to demonstrate the chaperone-like function of LONP1, the results are not convincing and the experiment requires more controls with individual mtHSP70 proteins (e.g. 70, JA3, EL1). In Fig. 4b, why is LONP1 protein in the pellet and not fully soluble?

Response: We examined the individual effect of DNAJA3 or GRPEL1 when combined with LONP1 S855A and mtHSP70 into the in vitro OXA1L reaction (Supplementary Fig. 5d). This analysis indicates that neither DNAJA3 nor GRPEL1 alone increases the solubility of OXA1L. Both have to be present with mtHSP70 to increase OXA1L solubility.

A fraction of LONP1 partitions to the pellet fraction when OXA1L is present in the reaction. This

is likely because insoluble OXA1L is bound to LONP1. This interpretation is supported by the observation that LONP1 is soluble when the control protein DHFR is present in the reaction (compare Supplementary Fig. 5b to Fig. 5b and Supplementary Fig. 5c). Therefore, the LONP1 is soluble unless in the presence of substantial substrate that fails to fold.

One wonders if unfolded subunits and/or misassembly of the LONP1 complex aggregate in the pellet fraction. In Fig. 4c, why is there less total OXAL1 protein with 1 uM and 2 uM S855A protease LONP1 mutant?

Response: As explained in detail in a response above to reviewer #1 (point 5), we can detect low levels of residual protease activity in the S855A mutant when very high protein concentrations are used. This activity may explain the slightly reduced levels of OXA1L when S855A is present (Fig. 5c).

In Fig. 4c, why is there less DNAJA3 short form in the samples with 1 uM and 2 uM LONP1 protease mutant S855A?

Response: As noted immediately above, the S855A mutant has low levels of residual protease activity. This likely explains the slightly reduced levels of DNAJA3 when S855A is present.

Line 200. The following statement requires further experimental support- "Addition of WT LONP1 resulted in degradation of OXA1L, likely because it is recognized as a misfolded substrate (Extended Data Fig. 4a)." To demonstrate that OXAL1 is degraded by LONP1, reactions should be incubated +/- ATP, show that loss of OXAL1 is ATP-dependent.

Response: The dependence of the degradation on ATP hydrolysis is demonstrated by analysis of the ATPase mutant LONP1(K529R). LONP1(K529R) does not affect the level of OXA1L, indicating that degradation is dependent on ATP hydrolysis activity (Fig. 5b). The *in vitro* translation reaction requires ATP to power ribosome-mediated translation. As a result, we cannot remove ATP from our reactions and must rely on mutants to show dependence on ATP hydrolysis.

Again, in Extended Data Fig. 4b, one wonders why there is LONP1 in the pellet fraction, and why there is more LONP1 in the pellet when CDDO is present. There is also more OXAL1 in the pellet fraction + CDDO. Therefore a control lane should be included showing the input proteins- S855A, mtHSP70, JA3 and EL1 before and after 1 hour at 37 degrees C.

Response: In the revision, Supplementary Fig. 5b shows that LONP1 is soluble in control reactions with *in vitro* translated DHFR. Therefore, LONP1 is normally soluble and becomes insoluble only when the substrate is highly insoluble. When CDDO is present, there is more insoluble OXA1L and correspondingly more insoluble LONP1.

In the experiments with CDDO (Supplementary Fig. 5b), LONP1 is separately preincubated with CDDO before its addition to the *in vitro* reaction. In control experiments, we analyzed LONP1 in the supernatant and pellet fractions before and after CDDO preincubation. The solubility of LONP1 was not affected by preincubation with CDDO (Figure below).

There are two significant caveats in this study. One caveat is that CDDO is not a LonP1-specific inhibitor (Figs. 1 and 2). The second is that doxycycline (DOX) has been shown to inhibit mitochondrial protein synthesis. Here, DOX is used to induce OXA1L-flag expression, in cells which may confound the interpretation of results. These limitations should be addressed, or alternatives approaches employed.

CDDO has multiple and diverse targets throughout the cell and also within mitochondria. For example, I-kappaB kinase, Jak1 and STAT3 are direct targets of CDDO (see references 1 and 2 at the end of this critique), in addition to other targets (references 3 and 4). In addition, the mechanism by which CDDO inhibits LonP1 has not been published, and it is likely that CDDO blocks other mitochondrial proteins. Therefore, results from experiments using CDDO are not clearly interpretable.

Response: We agree with the reviewer that CDDO is known to have multiple target proteins in the cell. However, we show that expression of LONP1 WT or S855A rescued CDDO-induced aggregation of OXA1L, NDUFA9, mtHSP70, and DNAJA3 (Supplementary Fig. 3a). These results indicate that the protein aggregations caused by CDDO were caused specifically by loss of LONP1 activity, not other targets. Additionally, our proteomic results show significant overlap of aggregated proteins from LONP1 knockdown and CDDO-treated mitochondria (Fig. 2b, d).

The issue with DOX is discussed below.

Doxycycline (DOX) has been shown to inhibit mitochondrial protein synthesis in cultured cells at a concentration of 1.0 microgram/ml, which is used in this study (reference 5). This effect of DOX is particularly relevant as OXA1L interacts with mitochondrial ribosomes, mediating the insertion of mitochondrial- as well as nuclear- encoded proteins into the mitochondrial inner membrane. Therefore, in experiments using DOX, an important control would be to determine the protein levels of mtDNA-encoded proteins. Control experiments are also crucial for determining the separate and combined effects of CDDO and DOX on the synthesis and stability of mtHSP70 and DNAJA3 as well as LonP1, OXA1L and NDUFA9.

Response: We agree with the reviewer that 1 µg/ml DOX treatment can disrupt mitochondrial protein synthesis and cause mitochondrial dysfunction. To avoid this harmful effect, we have reanalyzed DOX-inducible expression for OXA1L and mtHSP70 at varying DOX concentrations and found that 0.1 µg/ml DOX or lower is sufficient to reproduce the previous results. At 0.1 µg/ml DOX, there is no effect on expression of the mtDNA encoded protein MT-CO1 (Supplementary Fig. 1c), which is consistent with the literature^{2,3}. In the revised manuscript, 0.05 µg/ml DOX was used for LONP1 expression and 0.1 µg/ml DOX was used for OXA1L and mtHSP70 expression.

Minor comments:

Introduction

The Introduction is quite abbreviated. For the benefit of general readers, a brief description of the AAA+ LONP1 protease and its ATPase and proteases activities would be helpful. Also include a brief description of mtHSP70- an ATP-dependent chaperone, its co-chaperone DNAJA3, which stimulates the ATPase activity of mtHSP70 thereby releasing bound protein substrate, and the exchange factor GRPEL1, which stimulates the release hydrolyzes ADP allowing re-binding of ATP. Help for the readers to also briefly describing the localization and function OXA1L and NDUFA9 (e.g. caboxyl-terminus of this OXA1L interacts with mitochondrial ribosomes mediating insertion of both mitochondrial and nuclear encoded proteins into the mitochondrial inner membrane; and NDUFA9 subunit is at the membrane interface of Complex I NADH dehydrogenase, which binds NADH and required for complex assembly or stability).

Response: The revised manuscript now has a full introduction that discusses the topics raised by the reviewer.

Line 45. LONP1 has long been “suspected”. Perhaps “proposed” is a more accurate word than “suspected” as acting as a chaperone in the assembly of oxidative phosphorylation (OXPHOS) complexes in yeast and mammalian mitochondria. Suggest the following edit-
“LONP1 has been proposed to have a chaperone-like function in the assembly of oxidative phosphorylation (OXPHOS) complexes in yeast and mammalian mitochondria, which is independent of its protease activity.”

Response: We have modified the wording as suggested.

Results

Provide a line about why the study concentrates on OXA1L and NDUFA9.

Response: We have included the following statement. “We tested these proteins because they have important functions in mitochondrial biology. OXA1L is an insertase of the inner membrane, and NDUFA9 is a component of Complex I of the respiratory chain.”

Figure 1b. Although overexpressing WT and protease mutant LONP1 reduces the insoluble fraction of OXA1L and NDUFA9 considerably, they do not increase the soluble fraction compared to vector. Why might this be observed?

Response: In the revised manuscript, we use lower concentrations of DOX to induce WT LONP1 and the protease mutant at endogenous levels (Fig. 1b). Under these more optimal conditions, it is clear that LONP1 expression does increase the soluble fraction of OXA1L and NDUFA9 compared to vector.

Throughout the results, a good practice would be to include quantification of immunoblots and number of replicates performed.

Response: In the revised manuscript, we quantified immunoblots of the in vitro chaperone assay from three independent experiments (Fig. 5d, f).

Line 91 is unclear“Because LONP1 is required for proper OXA1L biogenesis, it is possible that LONP1 inhibition disrupts the function of chaperone systems in the mitochondrial matrix. We therefore tested the status of OXA1L upon disruption of HSP60 and mtHSP70”.]

Response: We have changed the wording to make the idea more clear: “Because depletion of LONP1 results in OXA1L and NDUFA9 aggregation, it is possible that LONP1 depletion disrupts the chaperone systems that facilitate protein folding in the mitochondrial matrix. We therefore tested the status of OXA1L upon disruption of HSP60 and mtHSP70 (also known as HSPA9), the two major chaperone systems of the matrix.”

Figure 2d- In LONP1 knockdown cells, overexpression of the LONP1 protease mutant S855A leads to decrease in total levels of DNAJA3, although the ratio of pellet: insoluble is less than vector control. Some comments about this would be helpful to the reader.

Response: In the revised manuscript, we optimize the rescue experiment by lowering the DOX concentration to give endogenous levels of the LONP1 protease mutant. Under such conditions, there is no decrease in the total levels of DNAJA3 (Fig. 3a).

=====

Reviewer #3 (Remarks to the Author):

This manuscript by Shin et al concerns the Lon protease of human mitochondria, LONP1. Specifically, the authors probe a role of Lon in promoting protein folding, making use of a variant lacking proteolytic activity. Overall the data presented in the manuscript makes a good case for interplay between LONP1 and the mitochondrial Hsp70 system in a prevention of aggregation/protein folding pathway. Thus, this manuscript could make an important contribution to the field, because although the idea that Lon has “chaperone-like” activity has been discussed, there has been little solid evidence to support it. However, there is a serious issue with the writing that leads to much confusion about how the authors are interpreting their data and what they are trying to say. In several cases, particularly in the “results” sections, terms (for example, “physical interaction”; “aggregation”) are used in a way that causes these problems. A serious rewriting is imperative.

Specific comments regarding experimental data:

1. Why is DNAJA3 not identified in the mass spec data, when it is so obvious in the western blots shown in fig 2a?

Response: In our mass spectrometry analysis, DNAJA3 is aggregated 1.3-fold higher in LONP1 knockdown mitochondria and 2.3-fold higher in CDDO-treated mitochondria compared to control (Supplementary Table 1). These results are consistent with the Western blot, but not as dramatic. A possible explanation is that DNAJA3 was abundant in the insoluble pellet from control mitochondria (45th in abundance out of 445 proteins in control sample). For the mass spectrometry procedure, the sample preparation takes much longer than for Western blot analysis. DNAJA3 may be aggregation-prone and the basal level of aggregation found under

the control condition may make the signal-to-noise less prominent in the mass spectrometry data compared to Western blotting. The basal insolubility of DNAJA3 in control mitochondria was verified by the Western blotting of isolated mitochondria below.

2. Related to comment 1, and as a more general comment for several experiments. It is important to show total as well as sup and pellet fractions as a control to confirm that close to 100% of the starting material is being recovered.

Response: In Supplementary Fig. 1a, we now examine the total, supernatant, and pellet fractions; the data show that our fractionation protocol results in no protein loss.

3. As a control, analysis of the effect of depletion of ClpX would be valuable. Presumably, the Hsp70 and DNAJA3 would not be present in aggregates and thus bolster the authors' conclusion that LONP1 is acting specifically in the Hsp70 pathway.

Response: This is a useful suggestion. In the human mitochondrial matrix, CLPP and CLPX form the AAA+ protease CLPXP. To test the effect of CLPXP on mitochondrial protein solubility, we generated CLPP knockdown and CLPX knockdown cells. Neither of the knockdown cells showed significant difference in the solubility of OXA1L or NDUFA9 (Supplementary Fig. 1b).

4. Since Hsp70 systems, as well as Lon, are dependent on ATP binding and hydrolysis, in vitro OXA1L solubility assay (Fig.4C in the presence of LONP1 and chaperones) should be done in the presence of excess of either ATP or ADP. The results should be more informative regarding the claims that the ADP state of Hsp70 matters and the ATPase of LONP1 also matters. Note: It is not clear in the M&M or figure legend what nucleotide was present in excess.

Response: It is a good suggestion to manipulate ATP/ADP levels in the in vitro assay, but this approach is not possible. The PURExpress in vitro translation system uses ATP as the energy source. It contains 2 mM ATP and a creatine kinase system for ATP recycling. We found that the addition of ATP or ADP into the reaction significantly disrupted the production of translation products. Nevertheless, we were able to address whether the chaperone-like activity of LONP1 is dependent on ATP hydrolysis. The ATP hydrolysis mutant K529R is unable to solubilize OXA1L (Fig. 5b) and CLPX (Supplementary Fig. 5g).

Specific major comments on writing.

5. A major cause of confusion in this manuscript is the use of the term "Hsp70 solubility" (as in

Line 122, the heading “Lon is required for mtHsp70 solubility”; also lines 128-132) for the case when Hsp70 is found in protein aggregates. For the first two reads through the results section this reviewer thought that the authors were trying to make the case that Lon was necessary to keep Hsp70 properly folded and active itself (that is was a chaperone for Hsp70). Using the same term for the case when a protein substrate of the chaperone system aggregates and when a chaperone is actively involved in trying to resolubilize aggregates/prevent aggregation should be avoided.

Response: To avoid the confusion raised by the reviewer, we have revised the wording. For example, the section heading has been changed to “Depletion of LONP1 results in mtHSP70 aggregation,” a more neutral statement that avoids the implications the reviewer indicated. The rest of the text has been similarly revised with this issue in mind.

6. Another cause of confusion is that the authors consider that anything that comes down in a pull-down is “physically interacting”. An example is Line 153, heading “Lon physically interacts with mtHsp70”. Because finding actual physical interactions (that is the direct physical interaction of one protein with another) between chaperone systems is a very active field of research, using this term of things coming down together in large amorphous aggregates is very confusing.

Response: We have revised the section and use the term “co-immunoprecipitation” instead of physical interaction. For example, the section heading has been revised to “LONP1 co-immunoprecipitates with mtHSP70.” In the subsequent text, similar changes have been made.

7. These and other problems could be helped by a suitable introduction to the Hsp70 system – how it works, what happens upon substrate aggregation, etc. A standard introduction would set the framework for how terms are being used in the results sections.

Response: A full-length introduction has been added to the manuscript and includes more information on the HSP70 system.

Minor points.

8. Line 111 -According to the source file with MS results OXA1L was 3.380921505- and 7.162547035 -fold increased - not 3.9.

Response: In our renewed analysis of triplicate mass spectrometric datasets, OXA1L aggregation is 3.0 and 5.8-fold elevated in LONP1 and DNAJA3 knockdown mitochondria, respectively. We have revised the text accordingly.

9. Line 114 – not clear where the statement that correlation coefficient equals 0.902 comes from; in Fig.2c on the graph it is $R^2 = 0.8139$.

Response: In the revised manuscript, the correlation coefficient R^2 of the aggregation profiles in LONP1 and DNAJA3 knockdown from our triplicate datasets is 0.6687. We have revised the text accordingly.

10. Line 125 - “Supplementary” Table instead of “Extended Data”

Response: We have corrected the text.

11. Line 130/132 -This sentence comes out of nowhere. The experimental approach, including use of previously unmentioned inhibitor, should be explained more.

Response: Because this experiment (Fig. 2e in the original submission) did not play an important role in the original submission and can be a source of confusion, we have decided to remove the data from the manuscript.

Our original intent was to show another example of mtHsp70 and DNAJA3 aggregation that could be restored by LONP1 (and LONP1 S855A) overexpression.

References

- 1 Ghosh, J. C. *et al.* Akt phosphorylation of mitochondrial Lonp1 protease enables oxidative metabolism and advanced tumor traits. *Oncogene* **38**, 6926-6939, doi:10.1038/s41388-019-0939-7 (2019).
- 2 Ahler, E. *et al.* Doxycycline alters metabolism and proliferation of human cell lines. *PloS one* **8**, e64561, doi:10.1371/journal.pone.0064561 (2013).
- 3 Moullan, N. *et al.* Tetracyclines Disturb Mitochondrial Function across Eukaryotic Models: A Call for Caution in Biomedical Research. *Cell reports* **10**, 1681-1691, doi:10.1016/j.celrep.2015.02.034 (2015).

REVIEWER COMMENTS

Reviewer #1 (Remarks to the Author):

The authors satisfactorily addressed all points raised on the previous version. I therefore support publication of this revised version in its present form.

Reviewer #2 (Remarks to the Author):

The manuscript revisions and responses to reviewer concerns are acceptable.

Reviewer #3 (Remarks to the Author):

The authors have addressed my comments/questions to my satisfaction.

Reviewer #4 (Remarks to the Author):

Reviewer Comments

“LONP1 and mtHSP70 cooperate to promote mitochondrial protein folding”

I was asked to review the proteomics portion of this paper, so my comments are related to this portion of the work.

1. In the experimental section, resolving power is provided for full scan (MS1), but not for MS2 scans. This information should be added. Otherwise, the description is good.
2. In my experience, the database search tolerance used for the parent ion, 10 ppm, is not optimal for full scans acquired at 60,000 resolving power. This would result in a smaller number of peptide assignments, which means that the actual number of protein IDs could be higher than reported. The searches should be performed with a tolerance of 20 ppm. For fragment ion tolerance, I cannot evaluate because the resolving power for these scans was not included in the acquisition parameters. I would suggest repeating the search with a broader tolerance. Doing so might give a higher number of quantifiable proteins.
3. In general, the information about statistical experimental design is lacking. Despite saying in the statistics section of the Reporting Summary that all information was included, it was not. In no particular order:
 - a. Were the samples biological replicates (3 separate biological experiments), technical replicates (multiple preparations from the same biological sample), or analytical replicates (3 injections of the same sample)? This information should be included.
 - b. In what order were the samples analyzed? Specifically, were they randomized? If so, this should be indicated. The most correct experimental design would have the samples run in a block-randomized fashion (3 analytical blocks with one sample from each condition per block, with the samples analyzed in a different order in each block). Complete randomization, while not optimal, is better than no randomization at all. If the same samples from each condition are run together (i.e. all control samples run first), then you cannot confidently say if differences are truly biological or if they are from instrument drift over time. If the samples were randomized or block randomized, a statement to this effect should be added.
 - c. Generally, the standard protocol is to do Benjamini-Hochberg multiple testing correction for quantitative proteomics data. Why was this not done? This correction needs to be performed.
4. The description of the protocol is good, but the reporting of proteomics data is incomplete. The standard practice is to report all peptide identifications, and no information is given regarding the number of peptide identifications for each protein. Moreover, protein identifications are also typically reported with the Uniprot accession number. A single gene symbol can belong to multiple protein

sequences within Uniprot. Supplementary Tables 1 and 2 give only gene symbols, fold changes and P-values with no other supporting information. I suggest adding columns for Uniprot accessions and also the number of peptide identifications per protein.

5. This paper does not follow acceptable proteomics publication guidelines regarding data availability; the statement given by the authors is completely unacceptable. For a study of this type, all data files and database search results should be uploaded to a public repository such as ProteomeXchange, to allow evaluation of the data. The authors should perform a complete upload of their mass spectrometry data to ProteomeXchange or a comparable public repository.

Reviewer Comments

“LONP1 and mtHSP70 cooperate to promote mitochondrial protein folding”

I was asked to review the proteomics portion of this paper, so my comments are related to this portion of the work.

1. In the experimental section, resolving power is provided for full scan (MS1), but not for MS2 scans. This information should be added. Otherwise, the description is good.

Thank you for pointing out this omission. The ms2 scans were collected at 30,000 resolution. This information has been added to the Methods description. All text changes in the revised manuscript have been highlighted in red.

2. In my experience, the database search tolerance used for the parent ion, 10 ppm, is not optimal for full scans acquired at 60,000 resolving power. This would result in a smaller number of peptide assignments, which means that the actual number of protein IDs could be higher than reported. The searches should be performed with a tolerance of 20 ppm. For fragment ion tolerance, I cannot evaluate because the resolving power for these scans was not included in the acquisition parameters. I would suggest repeating the search with a broader tolerance. Doing so might give a higher number of quantifiable proteins.

We thank the reviewer for this suggestion. However we respectfully disagree that 20 ppm should be used as the precursor mass tolerance. 10 ppm is the standard default precursor mass tolerance used for data from Orbitrap mass spectrometers with every common search algorithm and is not typically changed when different resolving powers are used for the ms1 scans. However, we agreed that it was worthwhile to look into this further.

First, we created a histogram of the mass errors of all peptide spectrum matches (PSMs) in the current data set, and calculated the average, standard deviation, and confidence levels. As shown below, the mass errors form a normal distribution with an average of 0.204 and a standard deviation of 1.637. Based on this, more than 99.7% of all identified spectra had mass errors less than 5.2 ppm, suggesting that few additional true positive identifications would result from increasing the precursor mass tolerance beyond 10 ppm.

To further explore this issue, we repeated the search with identical parameters except using a 20 ppm precursor mass tolerance. The overall identification numbers from both searches confirmed that few additional peptides are found, resulting in only 5 more proteins identified in total. Given that the observed increase in identifications was negligible and represents inclusion of poorer scoring PSMs which are much more likely to be false positives (the local FDR of the additional PSMs could be as high as 50%), we conclude that it is best to conform to the standard 10 ppm and leave the data set as reported rather than include a few extra but lower confidence proteins.

	10 ppm	20 ppm
Protein groups	4,000	4,005
Peptide groups	49,104	49,317
PSMs	235,123	235,852

3. In general, the information about statistical experimental design is lacking. Despite saying in the statistics section of the Reporting Summary that all information was included, it was not. In no particular order:

a. Were the samples biological replicates (3 separate biological experiments), technical replicates (multiple preparations from the same biological sample), or analytical replicates (3 injections of the same sample)? This information should be included.

The samples were biological replicates. This information has been added to the Methods section.

b. In what order were the samples analyzed? Specifically, were they randomized? If so, this should be indicated. The most correct experimental design would have the samples run in a block-randomized fashion (3 analytical blocks with one sample from each condition per block, with the samples analyzed in a different order in each block). Complete randomization, while not optimal, is better than no randomization at all. If the same samples from each condition are run together (i.e. all control samples run first), then you cannot confidently say if differences are truly biological or if they are from instrument drift over time. If the samples were randomized or block randomized, a statement to this effect should be added.

We thank the reviewer for this comment. It is true that instrument drift can lead to small changes in performance that may have effects on a small subset of identifications. This is especially of concern in the case of large clinical studies where dozens or hundreds of samples are analyzed over many days or even weeks. Such studies should include sample randomization as described along with proper quantitative controls like heavy spike-in standards for normalization. However, in small studies such as the current one, this is usually not a major concern.

For our data set, each set of WT, LONP1 kd, and DNAJA3 kd samples were prepared and analyzed at different times (although not intentionally randomized). Given how vastly different the mitochondrial protein profiles of these two sample sets are, we believe it better to analyze each set separately as carryover between samples would likely lead to much greater detrimental effects than instrument drift.

c. Generally, the standard protocol is to do Benjamini-Hochberg multiple testing correction for quantitative proteomics data. Why was this not done? This correction needs to be performed.

We apologize for the inadequate description of the statistics performed. Benjamini-Hochberg multiple testing correction was performed, and the reported p -values are the BH-adjusted p -values. We have updated the statistical description in the text to make this clear.

4. The description of the protocol is good, but the reporting of proteomics data is incomplete. The standard practice is to report all peptide identifications, and no information is given regarding the number of peptide identifications for each protein. Moreover, protein identifications are also typically reported with the Uniprot accession number. A single gene symbol can belong to multiple protein sequences within Uniprot. Supplementary Tables 1 and 2 give only gene symbols, fold changes and P-values with no other supporting information. I suggest adding columns for Uniprot accessions and also the number of peptide identifications per protein.

Thank you for the suggestions. We have added the requested information (Uniprot accession number, number of peptide identifications per protein) into the Supplementary Tables 1 and 2. We have also added Supplementary Tables 3 and 4, which list all proteins and peptides identified.

5. This paper does not follow acceptable proteomics publication guidelines regarding data availability; the statement given by the authors is completely unacceptable. For a study of this type, all data files and database search results should be uploaded to a public repository such as ProteomeXchange, to allow evaluation of the data. The authors should perform a complete upload of their mass spectrometry data to ProteomeXchange or a comparable public repository.

Data files have been uploaded to ProteomeXchange via the PRIDE database, to be made public upon manuscript publication, with the following access code: PXD021939.

Username: reviewer_pxd021939@ebi.ac.uk

Password: v3eq8siy

REVIEWERS' COMMENTS

Reviewer #4 (Remarks to the Author):

The authors have done an outstanding job of addressing the issues I raised, and I have no further suggestions. I think this is an excellent paper and is highly worthy of publication in Nature Communications. Congratulations to all the authors!

I liked the histogram of mass error distributions and found it to be enlightening. I typically acquire MS1 at 120K resolving power, and I think my mass error distribution might be slightly tighter than what you have. Otherwise, it's very similar. Well done!

Cheryl Lichti